# Euler-scale dynamical fluctuations
# in non-equilibrium interacting integrable systems

## Gabriele Perfetto[1][*] and Benjamin Doyon[2]

**1** SISSA – International School for Advanced Studies and INFN,
via Bonomea 265, 34136 Trieste, Italy
**2** Department of Mathematics, King's College London, Strand, London WC2R 2LS, U.K.

[*] gperfetto@sissa.it

## Abstract

We derive an exact formula for the scaled cumulant generating function of the time-integrated current associated to an arbitrary ballistically transported conserved charge. Our results rely on the Euler-scale description of interacting, many-body, integrable models out of equilibrium given by the generalized hydrodynamics, and on the large deviation theory. Crucially, our findings extend previous studies by accounting for inhomogeneous and dynamical initial states in interacting systems. We present exact expressions for the first three cumulants of the time-integrated current. Considering the non-interacting limit of our general expression for the scaled cumulant generating function, we further show that for the partitioning protocol initial state our result coincides with previous results of the literature. Given the universality of the generalized hydrodynamics, the expression obtained for the scaled cumulant generating function is applicable to any interacting integrable model obeying the hydrodynamic equations, both classical and quantum.

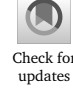

# 1  Introduction

One-dimensional isolated many-body quantum systems have recently received a great amount of attention thanks to ground-breaking advances in cold-atomic experiments, see e.g., Refs. [1, 2] for excellent reviews on the subject. In one dimension powerful analytical techniques are available for integrable models, which possess an extensive number of conservation laws, see e.g., the collection of reviews in Ref. [3]. In particular, it is by now well established, that integrable systems do not relax to a thermal steady state after an *homogeneous* quantum quench [4–6], where a global parameter of the Hamiltonian is abruptly changed and the system undergoes unitary evolution from a translationally invariant initial state. Relaxation occurs to the so-called *generalized Gibbs ensemble* (GGE), first proposed in Ref. [7] with Refs. [6, 8] reviews on the subject, which accounts for the presence of the infinite number of conservation laws in the thermodynamic limit.

Out of equilibrium, the initial state of the system is, however, often *inhomogeneous* and *non-stationary*. The study of the dynamics of interacting integrable models from inhomogeneous states is much more difficult than the one in homogeneous quantum quenches. In this context, the introduction of the *generalized hydrodynamics* (GHD), in the independent works in Ref. [9] and Ref. [10], has been a groundbreaking advancement. This theory is an extension of hydrodynamics to interacting integrable systems: while Euler hydrodynamics describes fluids with a finite number of conservation laws, the generalized hydrodynamics is concerned with integrable systems, admitting an infinite number of conserved quantities. As a consequence, the theory of generalized hydrodynamics is based on the GGE, instead of the canonical Gibbs ensemble. As GHD is a hydrodynamic theory, at its leading order, it applies in the so-called Euler-scaling limit, where the scale of space and time at which variations in the state of the system are observed, is taken to infinity. Under this assumption, at any space-time point $(x, t)$ the many-body system can be considered to locally relax to a GGE which depends on $(x, t)$, in the spirit of a local-equilibrium approximation [11]. The dynamics is ruled by hydrodynamic equations having a form analogous to the Euler equations. The input of the method is the thermodynamic Bethe ansatz (TBA), see, e.g., Refs. [12, 13]. Despite the name, the TBA is not restricted to quantum models: it applies also to classical systems, such as the hard-rod gas [14], the classical sinh-Gordon [15] and the Toda chain [16, 17]. The initial application of the GHD formalism in Refs. [9, 10], and later in Refs. [14, 15, 17–25], has been to the study of ballistic transport from the partitioning protocol initial inhomogeneous state, but the versatility of GHD allows to study various inhomogeneous setups such as the effect of confining potentials [26, 27], bump-release protocols [28, 29], correlation functions [30–32] and entanglement spreading [33–37]. Remarkably, the formalism can also be used to study non-integrable systems, provided the integrability breaking is weak enough so that on large enough length and time scales, the conserved charges may be assumed not to be broken [26, 38–47].

We also note that GHD has been experimentally verified in Ref. [48].

A complete understanding of the non-equilibrium processes where transport is present requires, however, the knowledge not only of the mean values of the ballistically transported conserved densities and the associated currents – the natural quantities at the basis of hydrodynamics – but also of the corresponding fluctuations. A relevant framework to account for fluctuations is given by the *large deviation theory*, see e.g., Refs. [49–51]. This framework builds on the computation of the large deviation function, which provides the statistics of rare but significant fluctuations. It generalizes to non-equilibrium conditions the concepts of entropy, thermodynamic phases and phase transitions, that are canonical quantities for describing the equilibrium statistical mechanics of many-body systems. The large deviation function can be computed from the associated scaled cumulant generating function (SCGF) [49], which is the non-equilibrium analogue of the free energy. In the context of isolated systems, until recently only few results for transport fluctuations were present, mostly regarding one-dimensional critical systems [52–54] and non-interacting models, such as the Levitov-Lesovik formula for free fermions [55–64], the free Klein-Gordon field theory [65] and harmonic chains [66]. Only recently, results became available in interacting integrable systems for the statistics of various observables not related to transport, see e.g., Refs. [67–70], and most importantly for the present paper, for ballistic transport [71,72]. In the latter, an exact expression for the current fluctuations at the Euler scale has been derived combining GHD and the large-deviation theory.

The results described above apply to homogeneous and stationary GGE states – even though these states can be obtained from inhomogeneous settings, such as the partitioning protocol for non-equilibrium steady states. At present, no general formula is available for the Euler-scale fluctuations of current flows within states affected by a nontrivial large-scale dynamics. The work [73] provided an important first step, obtaining, for non-interacting fermionic and bosonic systems, a formula for the scaled cumulant generating function describing the energy current fluctuations far from the connection point in the partitioning protocol, where a nontrivial Euler-scale dynamics occurs. The derivation is based on stationary phase methods, which are characteristic of free models, and therefore it is difficult to immediately generalize it to interacting integrable systems. A full understanding of the Euler-scale fluctuations of current flows in interacting integrable systems in states with large-scale inhomogeneity and dynamics is therefore still missing.

In this manuscript we aim at filling this gap, by providing an exact expression, in the Euler-scaling limit, for the SCGF of the time-integrated current associated to an arbitrary ballistically transported conserved charge. Our results apply to a broad class of inhomogeneous and dynamical states, which include, e.g., the initial state of the partitioning protocol, as shown in Fig. 1. The calculation of the SCGF is based on the biasing of the measure of the initial state by the exponential of the time-integrated current, in a similar way as the procedure followed in Refs. [71,72]. The biasing of the measure is shown to be fixed by the knowledge of two-point correlation functions in the inhomogeneous initial state, derived in Ref. [30]. Our derivation accordingly shows that the study of two-point correlation functions in GHD [30–32] is tightly related to the large-deviation theory of current fluctuations. The biasing technique developed in [72], generalised here to inhomogeneous, dynamical situations, is in principle applicable to a wide class of hydrodynamic systems, integrable or not. In this paper, however, we concentrate on integrable systems, where all the necessary technical tools are available for the technique to lead to calculable results.

We provide exact expressions for the first three cumulants of the time-integrated current and we further show that in the non-interacting limit, for the partitioning protocol initial state, our general formula for the SCGF reduces to the result of Ref. [73]. Our study of the SCGF thereby deeply generalizes the analysis of Refs. [71,72] and of Ref. [73] by accounting for inhomogeneous situations and interactions, respectively. The general expression presented for

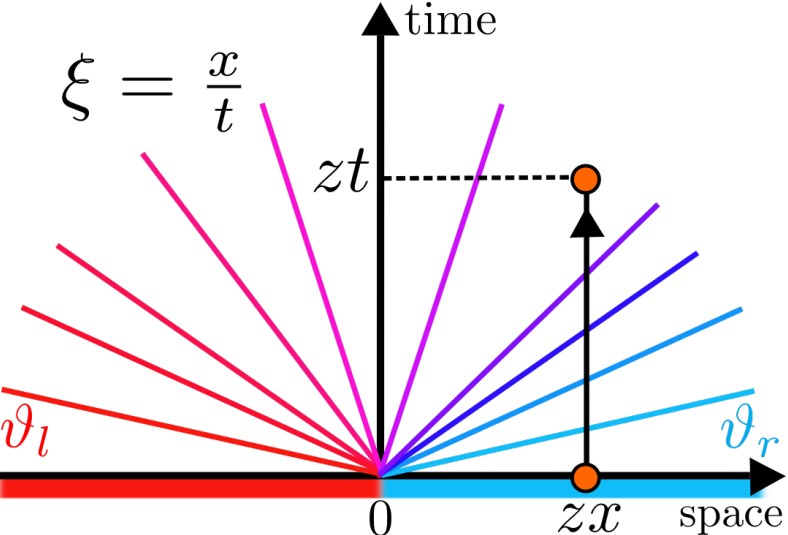

Figure 1: Schematic representation of the fluctuations of the time-integrated current in the Euler-scaling limit. In the Figure we consider, as an example, the partitioning protocol inhomogeneous state, where two GGEs, denoted as $\vartheta_l$ (in red at $x < 0$) and $\vartheta_r$ (in blue at $x > 0$), are joined at $x = 0$ at the time $t = 0$. The ensuing dynamics is described in terms of a continuum of states, constant on the light rays parametrized by $\xi = x/t$, which interpolate between $\vartheta_l$ and $\vartheta_r$. We consider the time integral $\Delta q_{i_*}(zx, zt) = \int_0^{zt} d\tau\, j_{i_*}(zx, \tau)$ of the current $j_{i_*}$, associated to the conserved charge $q_{i_*}$, flowing across the point $zx$ over the time interval $(0, zt)$. The fluctuations of $\Delta q_{i_*}(zx, zt)$ in the Euler-scaling limit are encoded in the SCGF $G(s, x, t) = \lim_{z \to \infty} 1/(zt) \ln \langle \exp(s\, \Delta q_{i_*}(zx, zt)) \rangle_{\text{inh},z}$. In the case depicted in the Figure, the average $\langle \ldots \rangle_{\text{inh},z}$ is taken over the partitioning protocol inhomogeneous state. We emphasize, however, that the formula for $G(s, x, t)$ derived in the manuscript applies more generally to a wide class of inhomogeneous and dynamical states varying at large scales $z$.

the SCGF relies solely on the large deviation theory and on the generalized hydrodynamics. It is therefore applicable to any integrable model described by the GHD, including the classical hard-rod gas [14, 32, 74, 75] and the Toda chain [16, 17], the classical sinh-Gordon field theory [15], the quantum Lieb-Liniger model [27–30], which describes the one-dimensional Bose gas, and quantum spin chains [9, 18–21, 23, 24], such as the XXZ model.

The rest of the paper is organized as follows. In Sec. 2 we review the basic concepts underlying the generalized hydrodynamics theory of integrable systems that will be needed for the presentation of our results. In Sec. 3 we briefly review the large-deviation theory for the rare fluctuations of ballistic current flows. Sec. 4 contains our main result for the exact expression of the SCGF in inhogeneous and dynamical initial states together with the applications of the result to the calculation of the cumulants and to the non-interacting limit. In Sec. 5 we draw our final conclusions. Some technical aspects are reported in the Appendix A.

## 2 Introduction to the generalized hydrodynamics

In this Section we review integrable models and their generalized hydrodynamics theory. The results that will follow in Sec. 4 apply to this class of systems. Reference [26] and the lecture notes in Ref. [76] provide a very good review on the subject. In Subsec. 2.1 we recall the basic identities regarding the description of the GGE in the thermodynamic Bethe ansatz formalism.

In Subsec. 2.2, we first define the Euler-scaling limit for a generic local observable, and then we report the main GHD hydrodynamic equations. In Sec. 2.3 we briefly review the expressions of the dynamical two-point functions in inhomogeneous and non-stationary states, derived in Ref. [30] and later numerically evaluated in Ref. [37], which constitutes a central building block of the results presented in Sec. 4.

## 2.1 The GGE and its thermodynamic Bethe ansatz description

We consider interacting integrable models in one spatial dimension, where, in the thermodynamic limit, an infinite number of local (or quasi-local) conserved charges $\{Q_i\}$ exists, with $i, j = 1, 2 \ldots \infty$. For Hamiltonian time evolution one of the conserved charges is the Hamiltonian of the system $H$; henceforth we further assume that the conserved charges commute among themselves $[Q_i, Q_j] = 0$, for any $i$ and $j$. The charges can be written as $Q_i = \int \mathrm{d}x\, q_i(x)$, where the density $q_i$ associated to $Q_i$ obeys a continuity equation with the current density $j_i$, corresponding to the current $J_i = \int \mathrm{d}x\, j_i(x)$, written in the form[1]

$$\partial_t q_i(x) + \partial_x j_i = 0, \quad \text{with} \quad i = 1, 2 \ldots \infty. \tag{1}$$

We will indicate by $q_0$, $q_1$ and $q_2$, respectively, the number, momentum and energy charges. In non-integrable models the latter are the only conserved quantities. The thermodynamic properties of integrable systems are described by the so-called *generalized Gibbs ensemble* (GGE) [6–8], which generalizes the canonical Gibbs ensemble to the case where an infinite number of conserved charges is present

$$\rho_{GGE} = \frac{e^{-W}}{Z_{GGE}}, \qquad \text{where} \qquad W = \sum_i \beta^i Q_i, \tag{2}$$

and $Z_{GGE}$ is the normalization constant. The parameters $\{\beta^i\}$ are Lagrange multipliers whose knowledge uniquely determines the GGE. Their values are fixed by the initial values of the conserved charges $\{Q_i\}$. In this manuscript we are using the superscript notation $\{\beta^i\}$ following the notation of Refs. [71, 72], but the superscript $i$ should not be confused with an exponent. Notice that the present discussion applies both to classical and quantum models. In the former case $\rho_{GGE}$ is a statistical distribution in the phase-space, while in the latter it is a density matrix.

The thermodynamic Bethe ansatz (TBA) formalism, see, e.g., Refs. [12, 77], provides an efficient method to represent the GGE. The system is described in terms of quasi-particles with rapidity $\lambda$, where each quasi-particle carries a bare charge $h_i(\lambda)$ corresponding to the single-particle eigenvalue of $Q_i$ in Eq. (1). The statistics of the quasi-particles is encoded into the free energy function $F(\varepsilon)$. The latter is given by $F(\varepsilon) = -\ln(1 + e^{-\varepsilon})$ for fermions, $F(\varepsilon) = \ln(1 - e^{-\varepsilon})$, for bosons, and $F(\varepsilon) = -e^{-\varepsilon}$ for classical particles. The GGE in Eq. (2) is then fixed by the pseudo-energy function $\varepsilon(\lambda)$, which, in turn, is determined by solving the non-linear integral equation

$$\varepsilon(\lambda) = w(\lambda) + \int \mathrm{d}\mu\, T(\lambda, \mu) F(\varepsilon(\mu)), \tag{3}$$

where $T(\lambda, \mu) = \Theta(\lambda, \mu)/2\pi$ and $\Theta(\lambda, \mu)$ is the differential two-body scattering phase. The term $w(\lambda)$ on the right hand side of Eq. (3) is named source term and it is defined as

$$w(\lambda) = \sum_i \beta^i h_i(\lambda), \tag{4}$$

---

[1] We use in this Section the continuum space notation just for simplicity, but the discussion carries over to discrete lattice models, e.g., spin chains.

where the $\{\beta^i\}$ are the parameters of the GGE in Eq. (2). For a Galilean model, for example, $h_0(\lambda) = 1$, $h_1(\lambda) = \lambda$ and $h_2(\lambda) = \lambda^2/2$. The only model-specic quantities in Eqs. (3) and (4) are the single-particle eigenvalues $h_i(\lambda)$, the integral kernel $T(\lambda, \mu)$, the free energy function $F(\varepsilon)$, and the spectral space $\int d\mu$. The latter, in the cases where multiple quasi-particle species $n$ are present, has to be enlarged to $\sum_n \int d\mu$. In this work we will drop the summation over the different quasi-particle species for brevity, but the results of Sec. 4 straightforwardly generalize to those cases.

From Eq. (3) it is customary to introduce the mode-occupation (or filling) function $\vartheta(\lambda)$, the root density $\rho_p(\lambda)$ and the state density $\rho_s(\lambda)$ as

$$\vartheta(\lambda) = \left.\frac{dF(\varepsilon)}{d\varepsilon}\right|_{\varepsilon(\lambda)}, \quad \rho_p(\lambda) = \vartheta(\lambda)\rho_s(\lambda), \quad \text{with} \quad \rho_s(\lambda) = \frac{1}{2\pi}1^{dr}, \tag{5}$$

where the dressing of a generic function $h^{dr}(\lambda)$ of $\lambda$ is defined by the linear integral equation

$$h^{dr}(\lambda) = h(\lambda) + \int d\mu\, T(\lambda, \mu)\vartheta(\mu)h^{dr}(\mu). \tag{6}$$

The GGE may be identified in various equivalent ways. The root density $\rho_p(\lambda)$ completely specifies it, as well as the mode occupation function $\vartheta(\lambda)$; since the knowledge of either determines the average values of all the conserved charges $\{Q_i\}$ of the model. As a consequence, in the following, we will the denote the average of a local observable $\mathcal{O}(x, t)$ at point $x$ and time $t$ over the GGE in Eq. (2) as

$$\langle \mathcal{O}(x, t)\rangle_\vartheta = \langle \mathcal{O}(0, 0)\rangle_\vartheta = \text{Tr}\left[\rho_{GGE}\,\mathcal{O}\right], \tag{7}$$

where the second equality follows from the fact that the GGE is homogeneous and stationary. The dependence of $\vartheta$ on $\lambda$ is omitted in this notation. In Eq. (7) we have used for convenience the quantum-mechanical trace notation, for classical systems it has to be replaced by an integral in phase space.

## 2.2 The Euler-scaling limit and the GHD equations

The TBA formalism and the GGE describe integrable systems in homogeneous and stationary states. The generalized hydrodynamics (GHD) theory accounts for situations where the state of the many-body system is inhomogeneous and non-stationary. In order to introduce this formalism, we first define the Euler-scaling limit, as the latter will be a central concept that will be used throughout the rest of the manuscript. Let us consider the case in which an integrable system is initialized in some generic state $\rho_0$ which is inhomogeneous and non-stationary. The average of a local operator $\mathcal{O}(x, t)$ over a generic inhomogeneous state $\rho_0$ will be denoted as $\langle \mathcal{O}(x, t)\rangle$. Upon considering inhomogeneities which slowly vary in space-time, one can use the local relaxation assumption or "local maximization of entropy principle" (an excellent book on this is Ref. [11]), which asserts that the system locally relaxes to a GGE:

$$\langle \mathcal{O}(x, t)\rangle \simeq \langle \mathcal{O}(0, 0)\rangle_{\vartheta(x, t)}. \tag{8}$$

The space-time dependence is recovered, in the spirit of a local-density approximation, by promoting the GGE Lagrange multipliers $\{\beta^i\} \to \{\beta^i(x, t)\}$, and, equivalently, the mode occupation function $\vartheta \to \vartheta(x, t)$, to be dependent on the space-time point $(x, t)$ where the operator is located. Equation (8) becomes exact in the limit of infinite length scale of the variation of the inhomogeneity. In the case of large, but finite, length scales for the variation in space and time of the inhomogeneity, instead, Eq. (8) is only approximate. The limit where Eq. (8) becomes exact is usually named "Euler-scaling limit" or simply "Euler scale". Henceforth we

will use both names interchangeably. For one-point functions at the space-time point $(x, t)$, we will use the subscript $\vartheta(x, t)$ to denote the averages over the local homogeneous GGE in Eq. (2) at $(x, t)$, as done in Eq. (8) consistently with the notation introduced in the previous Subsec. 2.1 in Eq. (7).

To make the discussion more concrete, here we introduce a particular class of inhomogeneous and dynamical initial states, which generalize the GGE in Eq. (2), which are given by

$$\rho_0 = \frac{1}{Z} \exp\left(-\sum_i \int_{\mathbb{R}} \mathrm{d}x\, \beta^i(x/z, 0)\, q_i(x, 0)\right). \tag{9}$$

In this expression $z$ is a scale parameter which has to be large enough such that the resulting Lagrange parameters $\{\beta^i(x/z)\}$ are functions of the position $x$ that vary only on large scales. If the multipliers $\{\beta^i\}$ do not depend on space, the state $\rho_0$ is a GGE as in Eq. (2). For this reason, we will refer henceforth to $\rho_0$ as *inhomogeneous* GGE.

In this manuscript, we will denote $\langle\dots\rangle_{\mathrm{inh},z}$ the averages over $\rho_0$ in Eq. (9). In the case of $\rho_0$ in Eq. (9) the Euler-scaling limit can be achieved by taking the limit where the scale of the inhomogeneity is sent to infinity $z \to \infty$, simultaneously with the space-time observation point $(x, t)$ of the operator $\mathcal{O}$. In formulas

$$\langle\mathcal{O}(0, 0)\rangle_{\vartheta(x, t)} = \lim_{z\to\infty} \langle\mathcal{O}(zx, zt)\rangle_{\mathrm{inh},z}$$

$$= \lim_{z\to\infty} \frac{1}{Z}\mathrm{Tr}\left[\mathcal{O}(zx, zt)\exp\left(-\sum_i \int_{\mathbb{R}} \mathrm{d}x\, \beta^i(x/z, 0)q_i(x, 0)\right)\right]. \tag{10}$$

A comment is in order here: in fact, sharp jumps in the state are also allowed and describable by Euler hydrodynamics. These are interpreted as "weak solutions" to the hydrodynamic equations. Therefore, $\beta^i(x, 0)$ do not need to be smooth functions: jumps are allowed, and the limit in Eq. (10) is still described by Euler hydrodynamics.

A particular initial inhomogeneity of the form of $\rho_0$ in Eq. (9), which we will discuss, is the partitioning protocol state, where indeed a sharp jump occurs. In this state two homogeneous GGEs, conventionally denoted as right ($r$ at $x > 0$) and left ($l$ at $x < 0$) are joined at the point $x = 0$ at time $t = 0$, see Fig. 1. Accordingly, the generalized inverse temperatures are chosen as

$$\beta^i(x, 0) = \beta_r^i\,\Theta(x) + \beta_l^i\,\Theta(-x), \tag{11}$$

where $\Theta(x)$ is the Heaviside step function. The initial filling function $\vartheta(x, 0, \lambda)$ corresponding to Eq. (11) is given by

$$\vartheta(x, 0, \lambda) = \vartheta_r(\lambda)\,\Theta(x) + \vartheta_l(\lambda)\,\Theta(-x), \tag{12}$$

where $\vartheta_{l,r}(\lambda)$ are fixed by the boundary conditions one imposes on the left $\{\beta_l^i\}$ and right $\{\beta_r^i\}$ reservoirs. For our results in Sec. 4 we will consider initial inhomogeneous and dynamical states of the form of Eq. (9) in the Eulerian limit $z \to \infty$.

Once the Euler-scaling limit in Eqs. (8) and (10) is assumed, it is rather simple to derive the hydrodynamic equations ruling the evolution of the system. Starting from Eq. (1), one has

$$\partial_t\langle q_i(0, 0)\rangle_{\vartheta(x, t)} + \partial_x\langle j_i(0, 0)\rangle_{\vartheta(x, t)} = 0. \tag{13}$$

The mean value of the charge density in Eq. (1) is given by the TBA description

$$\langle q_i(0, 0)\rangle_{\vartheta(x, t)} = \int \frac{\mathrm{d}\lambda}{2\pi} h_i(\lambda)\vartheta(x, t)1^{\mathrm{dr}}(x, t, \lambda), \tag{14}$$

where the dressing operation has been defined in Eqs. (5) and (6). In particular, all dressed quantities become functions of space and time, being determined by $\vartheta(x,t)$ via Eq. (6). The expression of the current in Eq. (1) has been first discovered in Refs. [9,10] and it reads as

$$\langle j_i(0,0)\rangle_{\vartheta(x,t)} = \int \frac{d\lambda}{2\pi}(E')^{dr}(x,t,\lambda)\vartheta(x,t)h_i(\lambda),\qquad(15)$$

where $h_2(\lambda) = E(\lambda)$ is the bare single-particle energy eigenvalue and the effective velocity $v^{eff}$ is given by

$$v^{eff}(x,t,\lambda) = \frac{(E')^{dr}(x,t,\lambda)}{1^{dr}(x,t,\lambda)} = \frac{(E')^{dr}(x,t,\lambda)}{(P')^{dr}(x,t,\lambda)},\qquad(16)$$

where in the last equality we used that for non-relativistic models $h_1(\lambda) = P(\lambda) = \lambda$ (even though the second relation in Eq. (16) holds also for more general parametrizations of the momentum as a function of the rapidity). The effective velocity $v^{eff}$ is a generalization of the group velocity $v_g$, which takes into account the interactions among the quasi-particles via the dressing operation. $v^{eff}$ has been first defined in Ref. [78] in the context of the light-cone spreading of correlations in homogeneous quantum quenches. We stress that Eq. (15) goes beyond the results available from the TBA and it has been first proved in relativistically invariant quantum field theories in Ref. [10] and numerically tested in quantum spin chains in Ref. [9]. Later, it has been proved in a variety of contexts including quantum spin chains and classical models [79–85]. Inserting Eqs. (14) and (15) into the continuity equation (13), and assuming the completeness of the set of local (or quasi-local) charges $h_i(\lambda)$, the final GHD equation for the filling function, see Refs. [9,10] for the original derivation, turns out to be

$$\partial_t\vartheta(x,t,\lambda) + v^{eff}(x,t,\lambda)\partial_x\vartheta(x,t,\lambda) = 0.\qquad(17)$$

Equation (17) express the fact that the occupation function $\vartheta(x,t,\lambda)$ represents the normal modes of the hydrodynamics, which, in integrable systems, form a continuum labelled by the rapidity $\lambda$. At the Euler scale, the normal modes are convectively transported with velocity $v^{eff}(x,t,\lambda)$. Generalizations of Eq. (17) to account for the presence of trapping potentials [26], diffusive corrections [43,86–89], space-time variations of the interaction terms of the Hamiltonian [38] and Markovian coupling to an external bath [40] have been further developed. For the derivation of our results in Sec. 4 the form in Eq. (17) will be sufficient.

As a final piece of introduction, we mention that Eq. (17) admits a solution by the characteristic function, $\mathcal{U}(x,t,\lambda,t_0)$, encoding the position at time $t_0$ of the characteristic curve $x(\mathcal{U},t,\lambda,t_0)$ of the quasi-particle with rapidity $\lambda$ that at time $t$ is in $x$ (therefore $\mathcal{U}(x,t_0,\lambda,t_0) = x$). The characteristic curve $x(\mathcal{U},t,\lambda,t_0)$ is defined as the curve tangent to the effective velocity $v^{eff}$ in Eq. (16). The function $\mathcal{U}(x,t,\lambda,t_0)$ is defined by inverting $x(\mathcal{U},t,\lambda,t_0)$ with the respect to the initial position $\mathcal{U}$, see Refs. [29,36] for a detailed discussion. From Eq. (17), it is simple to see that the filling function $\vartheta(x,t,\lambda)$ is constant along the characteristic $\mathcal{U}(x,t,\lambda,t_0)$

$$\vartheta(x,t,\lambda) = \vartheta(\mathcal{U}(x,t,\lambda,t_0),t_0,\lambda).\qquad(18)$$

Remarkably, in Ref. [29], it has been proved that $\mathcal{U}(x,t,\lambda,t_0)$ can be determined by solving the following integral equation

$$\int_{x_0}^{x}dy\,\rho_s(y,t,\lambda) = \int_{x_0}^{\mathcal{U}(x,t,\lambda,t_0)}dy\,\rho_s(y,t_0,\lambda) + v^{eff}(x_0,t_0,\lambda)\rho_s(x_0,t_0,\lambda)(t-t_0),\qquad(19)$$

where $x_0$ is an asymptotically stationary point (see the discussion after Eq. (83) in Appendix A). Equations (18) and (19) provide an efficient way to solve the main GHD equation (17) initial value problem, as shown in Ref. [29]. The integral equation (19) for $\mathcal{U}(x,t,\lambda,t_0)$, furthermore, is fundamental for the derivation of the exact expression of the Euler-scale dynamical correlation functions, which will be the object of the next Subsection.

## 2.3 Euler-scale dynamical correlation functions

The discussion of Subsec. 2.2 focused on the Euler scaling of one-point functions (mean values) of local observables. Since for weak inhomogeneities the system is composed by locally homogeneous space-time fluid cells, one-point functions at the space-time point $(x,t)$ can be computed based on the knowledge of their expression $\langle \mathcal{O}(0,0)\rangle_{\vartheta(x,t)}$ in the local homogeneous GGE in Eq. (2) at $(x,t)$ according to Eq. (10). Euler-scale connected correlation functions are, instead, harder to compute as they depend on the whole inhomogeneous state $\rho_0$ characterizing the initial state of the system, and not only on the homogeneous GGE on a specific fluid cell. Considering two-point functions at the space-time points $(0,0)$ and $(x,t)$, the approach for their calculation [11, 32, 90] consists in studying the linear response of the system at the space-time point $(x,t)$ to a local perturbation at the point $(0,0)$. Correlations, at the Euler scale, are then determined by the stable normal modes propagating between the two space-time points. In this way, exact expressions for the two-point functions of generic local observables in the large-scale limit can be obtained. However, the expressions obtained in this way in Refs. [11, 32, 90] assume that background fluid state at the initial space-time point $(0,0)$ is homogeneous and stationary. In Ref. [30], this approach has been extended to account for two-point correlation functions from inhomogeneous initial states of the form in Eq. (9), under homogeneous evolution (for a dynamics with an homogeneous and time-independent Hamiltonian – that is, without spatially or temporally dependent external force fields). We report just the main ideas behind the derivation together with the final expression of the the dynamical two-point function, which will be essential for the derivation of the scaled cumulant generating function detailed in Sec. 4.

Consider a small perturbation of one Lagrange parameter $\beta_j(y,0) \to \beta_j(y,0)+\delta\beta_j(y,0)$ in the initial state $\rho_0$ in Eq. (9). The response of the system to the small perturbation of $\beta_j(y,0)$ is related to the connected correlation function involving the associated density $q_j(y,0)$. Consider the average of some other density $q_i(x,t)$ over $\rho_0$ in Eq. (9): the functional derivative of this average with respect to $\beta_i(y,0)$ is

$$-\frac{\delta}{\delta\beta_j(y,0)}\langle q_i(x,t)\rangle_{\vartheta_0}^{\text{Eul}} = \langle q_i(x,t)q_j(y,0)\rangle_{\vartheta_0}^{c,\text{Eul}}, \tag{20}$$

with the Euler-scaling limit for the connected correlator defined as

$$\langle q_i(x,t)q_j(y,0)\rangle_{\vartheta_0}^{c,\text{Eul}} = \lim_{z\to\infty} z\Big(\big\langle q_i(zx,zt)q_j(zy,0)\big\rangle_{\text{inh},z} - \big\langle q_i(zx,zt)\big\rangle_{\text{inh},z}\big\langle q_j(zy,0)\big\rangle_{\text{inh},z}\Big). \tag{21}$$

In $\langle q_i(x,t)q_j(y,0)\rangle_{\vartheta_0}^{c,\text{Eul}}$ the subscript $\vartheta_0$ denotes the filling function which characterizes globally, as a function of space, the inhomogeneous state $\rho_0$ in Eq. (9) at the time slice $t=0$ in the Euler-scaling limit $z\to\infty$. For the partitioning protocol initial state, for example, $\vartheta_0$ is given in Eq. (12). This notation, where time appears as a lower index, stresses the fact that two-point correlation functions depend on the whole initial inhomogeneous state $\vartheta_0$ of the system and not only on the homogeneous GGE $\vartheta(x,t)$ at a specific space-time fluid cell $(x,t)$. For one-point functions, instead, we have denoted with $\langle q_i(x,t)\rangle_{\vartheta_0}^{\text{Eul}} = \langle q_i(0,0)\rangle_{\vartheta(x,t)}$ the average over $\rho_0$ in Eq. (9) in the limit $z\to\infty$ according to Eq. (10) and in agreement with the notation introduced in Subsecs. 2.1 and 2.2. [One-point functions of the charge densities and of the associated currents are given in Eqs. (14) and (15), respectively.]

We emphasize that, for some specific models, the Euler-scale limit of two-point correlation functions requires additional care and $q_i(zx,zt)$, $q_j(zy,0)$ in the r.h.s. of Eq. (21) have to be averaged over space-time fluid-cells, as explained in Ref. [30] and shown in Ref. [15] for the classical sinh-Gordon field theory. For the classical hard-rod gas considered in Ref. [37], instead, fluid-cell averaging is not necessary and the Euler-scale predictions for the two-point function are simply obtained by averaging in space.

Equation (20) represents an extension of the fluctuation-dissipation theorem of statistical mechanics [91] to the framework of GHD, where infinitely many conserved charges are present. From Eq. (20) an expression for the two-point connected correlation function $\langle q_i(x,t)q_j(y,0)\rangle^{c,\text{Eul}}_{\vartheta_0}$ can then be derived exploiting the fact that the expression of the one-point function on the l.h.s is known in Eq. (14), as shown in Ref. [30].

These results for correlation functions of conserved densities can then be extended to the connected two-point function of any pair of local observables $\langle \mathcal{O}(x,t)\mathcal{O}'(y,0)\rangle^{c,\text{Eul}}_{\vartheta_0}$, via hydrodynamic projections [32]. The main idea of the latter is that at the Euler scale correlations are determined by the ballistically propagating conserved charges. Correlations can then be determined by "expanding" the operator $\mathcal{O}(x,t)$ of interest in the basis of the local (or quasi-local) conserved charges

$$\mathcal{O}(x,t) = \sum_{j,k} q_j(x,t)\left(C_{[\vartheta(x,t)]}\right)^{-1}_{jk}\langle q_k|\mathcal{O}\rangle_{\vartheta(x,t)}, \tag{22}$$

where the scalar product $\langle q_k|\mathcal{O}\rangle_{\vartheta(x,t)}$ is computed at the space-time point $(x,t)$ of the observable as

$$\langle \mathcal{O}|q_k\rangle_{\vartheta(x,t)} = -\frac{\partial}{\partial \beta_k(x,t)}\langle \mathcal{O}\rangle_{\vartheta(x,t)} = \int d\lambda\, \rho_p(x,t,\lambda)f(x,t,\lambda)V^{\mathcal{O}}(x,t,\lambda)h^{\text{dr}}_k(x,t,\lambda). \tag{23}$$

$f(x,t,\lambda)$ is dubbed statistical factor of the model. For models with fermionic quasi-particle statistics $f(x,t,\lambda) = 1-\vartheta(x,t,\lambda)$, for bosonicic quasi-particles $f(x,t,\lambda) = 1+\vartheta(x,t,\lambda)$, while for classical particle models $f(x,t,\lambda) = 1$. $V^{\mathcal{O}}$ in Eq. (23) is named one-particle-hole form factor of the operator $\mathcal{O}$, it a functional of the filling function defined such that Eq. (23) holds. It must be worked out for every operator individually. For charge densities and the associated currents they are simply related to dressed single-particle eigenvalues $h^{\text{dr}}_i$ in Eq. (6) [32] as

$$V^{q_i} = h^{\text{dr}}_i \quad \text{and} \quad V^{j_i} = v^{\text{eff}}h^{\text{dr}}_i. \tag{24}$$

$C^{-1}_{\vartheta(x,t)}$ in Eq. (23) is the inverse of the correlation matrix (cf. Eq. (23) and Eq. (24) for $V^{q_i}$)

$$(C_{\vartheta(x,t)})_{ij} = \langle q_i|q_j\rangle_{\vartheta(x,t)} = \int d\lambda\, \rho_p(x,t,\lambda)f(x,t,\lambda)h^{\text{dr}}_i(x,t,\lambda)h^{\text{dr}}_j(x,t,\lambda). \tag{25}$$

The factor $C^{-1}_{\vartheta(x,t)}$ is introduced in Eq. (22) because the densities $\{q_i\}$ are not orthonormal under the scalar product $(C_{\vartheta(x,t)})_{ij} = \langle q_i|q_j\rangle_{\vartheta(x,t)}$ given in Eqs. (22) and (23). A detailed justification of the hydrodynamic projection identity in Eq. (22) is provided in Refs. [32, 92].

For our purpose, it is sufficient to say that an exact formula for two-point correlations of generic local observables $\langle \mathcal{O}(x,t)\mathcal{O}'(y,0)\rangle^{c,\text{Eul}}_{\vartheta_0}$ can be derived using Eq. (22) and the result for the two-point correlator of charge densities obtained from Eq. (20). The formula reads as

$$\langle \mathcal{O}(x,t)\mathcal{O}'(y,0)\rangle^{c,\text{Eul}}_{\vartheta_0} = \int d\lambda\, \rho_p(x,t,\lambda)f(x,t,\lambda)V^{\mathcal{O}}(x,t,\lambda)\left[\Gamma_{(y,0)\to(x,t)}V^{\mathcal{O}'}(y,0)\right](\lambda). \tag{26}$$

The square brackets in Eq. (26) denote the integral-operator notation

$$\left[\Gamma_{(y,0)\to(x,t)}h\right](\lambda) = \int d\mu\, \Gamma_{(y,0)\to(x,t)}(\lambda,\mu)h(\mu). \tag{27}$$

The propagator, $\Gamma_{(y,0)\to(x,t)}(\lambda,\mu)$, describes how the local perturbation at the space-time point $(y,0)$ is transported by the normal modes of the hydrodynamics on given characteristic curve $\mathcal{U}(x,y,\lambda,0)$, until it reaches the space-time point $(x,t)$. In particular, $\Gamma_{(y,0)\to(x,t)}$ can be split into two parts

$$\Gamma_{(y,0)\to(x,t)}(\lambda,\mu) = \delta(y - \mathcal{U}(x,t,\lambda,0))\,\delta(\lambda - \mu) + \Delta_{(y,0)\to(x,t)}(\lambda,\mu). \tag{28}$$

The first term is dubbed *direct* propagator, it describes the normal-modes propagation along the curved characteristic curve $\mathcal{U}(x,t,\lambda,0)$ (see Eqs. (18) and (19) in Subsec. 2.2) within the inhomogeneous fluid background. Therefore, only the quasi-particles with the suitable rapidity $\lambda$ to propagate from $(y,0)$ to $(x,t)$ enter in this term. On the other hand, $\Delta_{(y,0)\to(x,t)}$ is named *indirect* propagator and it encodes a more subtle effect. $\Delta_{(y,0)\to(x,t)}$ describes the perturbation to the trajectory of the rapidity $\lambda$ due to the interaction with other rapidities $\lambda'$. Quasi-particles with arbitrary rapidities $\lambda'$, not necessarily connecting the two observation points $(y,0)$ and $(x,t)$, are therefore involved into the definition of the indirect propagator. Inserting Eq. (28) into Eq. (26) an analogous decomposition for the two-point correlation formula applies

$$
\begin{aligned}
\left\langle \mathcal{O}(x,t)\mathcal{O}'(y,0)\right\rangle_{\vartheta_0}^{c,\mathrm{Eul}} &= \sum_{\gamma\in\lambda_\star(x,t,y,0)} \frac{\rho_s(x,t,\gamma)\vartheta(y,0,\gamma)f(y,0,\gamma)}{|\partial_\lambda\mathcal{U}(x,t,\gamma,0)|}V^{\mathcal{O}}(x,t,\gamma)V^{\mathcal{O}'}(y,0,\gamma) \\
&+ \int \mathrm{d}\lambda\,\rho_p(x,t,\lambda)f(x,t,\lambda)V^{\mathcal{O}}(x,t,\lambda)\Big[\Delta_{(y,0)\to(x,t)}V^{\mathcal{O}'}(x,t)\Big](\lambda),
\end{aligned}
\tag{29}
$$

where the term on the first line of the r.h.s of Eq. (29) is named *direct correlator*, while the term on the second line is dubbed *indirect correlator*. The set $\lambda_\star(x,t,y,0) = \{\lambda : \mathcal{U}(x,t,\lambda,0) = y\}$ over which the sum on the first line runs expresses the fact that direct correlations are determined solely by the propagation of the quasi-particles with the right rapidity $\lambda_\star$ to connect the space-time point $(y,0)$ with $(x,t)$.

Note that in Eqs. (26)-(29) we have placed for convenience the operator $\mathcal{O}'(y,0)$ at the space-time point $(y,0)$. However, the very same formulas apply for the operator $\mathcal{O}'(y,t_0)$ at a generic time $t_0 \neq 0$ upon replacing $\mathcal{U}(x,t,\lambda,0)$ with $\mathcal{U}(x,t,\lambda,t_0)$, according to Eqs. (18) and (19), and similarly for all TBA quantities evaluated at the time 0. In particular, Eqs. (26)-(29) apply also in the case $t < 0$, where the operator $\mathcal{O}(x,t)$ is computed at a time earlier than $\mathcal{O}'(y,0)$, and time ordering is not needed. This is a consequence of the fact that the Euler-scale equation (17), on which Eqs. (26)-(29) are based, is time reversible. The indirect propagator $\Delta_{(y,0)\to(x,t)}$ can be obtained by solving a linear integral equation, which we report in the Appendix A for completeness.

Only very recently, in Ref. [37], $\Delta_{(y,0)\to(x,t)}$ and the formula in Eq. (29) have been numerically evaluated for various interacting integrable models (Lieb-Liniger, sinh-Gordon and the classical hard-rod gas) in different non-equilibrium protocols involving inhomogeneous initial states. Moreover, in Ref. [37], the first numerical demonstration of the validity of the Euler-scale formulas in Eq. (29) has been given by comparing them with Monte Carlo simulations of the microscopic hard-rod dynamics. An excellent agreement has been, in particular, found for long times and large values of the scale parameter $z$ of the inhomogeneity in Eq. (9), consistently with the expectation that the Euler-scale predictions apply to weakly inhomogeneous settings.

## 3 Large deviation theory of ballistic transport

The large deviation theory, together with the generalized hydrodynamics, forms the basis of the results presented in Sec. 4 for the calculation of the SCGF in inhomogeneous initial states. In Subsec. 3.1, we accordingly briefly discuss the large deviation theory of ballistically transported conserved quantities. Then, in Subsec. 3.2, we briefly recall the results from Refs. [71, 72], which will be important for comparison with our analysis.

## 3.1 Large deviation principle and the scaled cumulant generating function

Considering the continuity equation in Eq. (1), each conserved density satisfies, for a particular density $q_{i_*}$ denoted by the subscript $i_*$ we define the time-integrated current

$$\Delta q_{i_*}(x,t) = \int_0^t d\tau \, j_{i_*}(x,\tau). \tag{30}$$

$q_{i_*}$ could denote, for example, the energy, the particle or any other density of the model. The GHD equation in Eq. (17) describes ballistic transport of the hydrodynamic modes. Ballistic transport, indeed, is relevant in integrable models as a consequence of the presence of an infinite number of conserved charges. For ballistic motion, the time-integrated current is expected to depend *extensively* on the time $t$ as $\Delta q_{i_*}(x,t) \propto t$ for large times $t$, and it is therefore convenient to focus on the *intensive* variable $J_{i_*} = \Delta q_{i_*}(x,t)/t$. According to the large deviation theory, see, e.g., Refs. [49–51], the probability density function of $J_{i_*}$ for large times $t$ peaks exponentially around the mean, and most-probable, value $\langle J_{i_*} \rangle$ as

$$p(J_{i_*}) \asymp \exp(-t I(J_{i_*})). \tag{31}$$

In Eq. (31) the notation "$\asymp$", taken from Ref. [49], denotes equality of the right and the left hand side in logarithmic scale in the long-time limit $t \to \infty$. Equation (31) is named large-deviation principle and it expresses the leading asymptotic dependence on $t$ of the probability density $p(J_{i_*})$. The function $I(J_{i_*})$ is dubbed large-deviation or rate function; it is convex, non-negative, with a unique zero around the average and most probable value $I(J_{i_*} = \langle J_{i_*} \rangle) = 0$, implying that large fluctuations of $J_{i_*}$ far from the mean value $\langle J_{i_*} \rangle$ are exponentially suppressed according to Eq. (31). The Legendre-Fenchel transform [49] of $I(J_{i_*})$ is the scaled cumulant generating function (SCGF) $G(s,x,t)$ of the time-integrated current in Eq. (30).

Here, as we are interested in the Euler-scaling limit, the SCGF is defined as

$$G(s,x,t) = \lim_{z\to\infty} \frac{1}{zt} \ln \langle \exp(s \, \Delta q_{i_*}(zx,zt)) \rangle_{\text{inh},z} = \sum_{k=1}^{\infty} \frac{s^k}{k!} c_k(x,t). \tag{32}$$

The average is taken over the rescaled inhomogeneous state $\rho_0$ in Eq. (9). Equation (32) serves to define the meaning of the probability $p(J_{i_*})$ in (31), which depends on the initial state, and therefore in particular on $z$; it is the scaling in $z$ as per Eq. (32) that gives rise to the asymptotic equality represented by "$\asymp$" in Eq. (31).

From the knowledge of $G(\lambda, x, t)$ in Eq. (32), one can obtain by Taylor expansion the cumulants $\{c_k\}$ of the time-integrated current, which are defined as

$$c_k(x,t) = \lim_{z\to\infty} \frac{1}{zt} \langle [\Delta q_{i_*}(zx,zt)]^k \rangle_{\text{inh},z}^c = \frac{1}{t} \int_0^t dt_1 \cdots \int_0^t dt_k \, \langle j_{i_*}(x,t_1) \dots j_{i_*}(x,t_k) \rangle_{\vartheta_0}^{c,\text{Eul}}, \tag{33}$$

where the Euler-scaling limit of the $k$-point connected correlation function $\langle \mathcal{O}(x_1,t_1) \dots \mathcal{O}(x_k,t_k) \rangle_{\vartheta_0}^{c,\text{Eul}}$ of a local observable $\mathcal{O}$ is defined in a similar way as in Eq. (21) for the two-point function [30][2]:

$$\langle \mathcal{O}(x_1,t_1) \dots \mathcal{O}(x_k,t_k) \rangle_{\vartheta_0}^{c,\text{Eul}} = \lim_{z\to\infty} z^{k-1} \langle \mathcal{O}(zx_1,zt_1) \dots \mathcal{O}(zx_k,zt_k) \rangle_{\text{inh},z}^c. \tag{34}$$

The validity of the large deviation principle in Eq. (31) implies that all that the connected correlation functions $\{\langle [\Delta q_{i_*}(zx,zt)]^k \rangle_{\text{inh},z}^c\}$ scale as $z$, with the cumulants $\{c_k\}$ being therefore finite and the series expansion in Eq. (32) being valid in some interval of $s$ around the

---

[2]As in the case of two-point functions, at the Euler scale, time-ordering is not necessary in the definition of $k$-point connected correlations. Fluid-cell averaging might, instead, be needed in Eq. (34), as discussed after Eq. (21).

origin. The main result of this manuscript is the derivation of an exact expression for $G(s, x, t)$ in Eq. (32) valid for interacting integrable models, both classical and quantum, which are described by GHD.

## 3.2 SCGF for homogeneous GGEs: review of the result

In Refs. [71,72] a general approach has been developed to compute $G(s, x, t)$ in the case where the average in Eq. (32) is taken w.r.t. the *homogeneous* GGE $\vartheta$ as in Eqs. (2) and (7) (and not w.r.t. the inhomogeneous state $\rho_0$ in Eq. (9), as it will be the case in Sec. 4). In this case, because of the homogeneity of the GGE, $G(s, x, t) = G(s)$. Let us review this approach.

The analysis of Refs. [71,72] relies on the GHD description of the Euler-scale hydrodynamics. In particular, we need to introduce the so-called *flux jacobian* matrix, $A_i^j$

$$A_i^j = \frac{\partial \langle j_i \rangle}{\partial \langle q_j \rangle}, \tag{35}$$

with $i$ the row index and $j$ the column one, which will play an important role in the analysis of this Subsection. This matrix describes how the average currents depends on the average densities and therefore it depends on the equation of state of the model. In the case of integrable systems, where the averages in Eq. (35) are computed over the homogeneous GGE $\vartheta$ in Eqs. (2) and (7) as per Eqs. (14) and (15), an expression for the matrix elements of $A_i^j$ can be given by means of the hydrodynamic projection formalism, as shown in Ref. [32]. In the basis of the single-particle eigenvalues $\{h_i(\lambda)\}$ the matrix $A_i^j$ reads

$$A_i^j = \int d\lambda\, h_i^{dr}(\lambda) v^{eff}(\lambda) h_{dr}^j(\lambda), \tag{36}$$

with the effective velocity $v^{eff}(\lambda)$ given in Eq. (16). The function $h_{dr}^i(\lambda)$ denotes the orthonormal conjugate of $h_i^{dr}(\lambda)$, which satisfies the orthogonality and completeness relations

$$\int d\lambda\, h_{dr}^j(\lambda) h_i^{dr}(\lambda) = \delta_i^j,$$
$$\sum_j h_{dr}^j(\lambda) h_j^{dr}(\lambda') = \delta(\lambda - \lambda'), \tag{37}$$

respectively. Notice that the matrix $A_i^j$ in integrable systems is infinite-dimensional, yet it can be formally defined. From Eqs. (36) and (37) the eigenvalue equation for $A_i^j$ readily follows as

$$\sum_j A_i^j h_j^{dr}(\lambda) = v^{eff}(\lambda) h_i^{dr}(\lambda), \tag{38}$$

which shows that the flux jacobian has a continuous spectrum indexed by the rapidity $\lambda$ with eigenvalue the effective velocity $v^{eff}(\lambda)$.

In order to compute $G(s)$ one then biases the GGE measure in Eq. (2) by multiplying it by the exponential of the time-integrated current appearing in Eq. (32). Averages over this tilted measure become dependent on the parameter $s$ conjugate to the time-integrated current $\langle \mathcal{O} \rangle_{\vartheta(s)}$. Note that we are here extending the notation introduced in Eq. (7) for the average over the homogeneous GGE by promoting $\vartheta(s)$ to be dependent on $s$ (in addition to the rapidity $\lambda$ whose dependence is not reported for brevity). In Subsec. 4.2 we will explain this procedure in more details, for the moment we just report the main result for the *flow equation* from Ref. [71,72].

The flow equation describes how the homogeneous GGE $\vartheta$ in Eq. (2) is modified by the insertion of the exponential of the time-integrated current. Fundamentally, the $s$ modified state $\vartheta(s)$ is still an homogeneous GGE, yet with Lagrange parameters $\{\beta^n\}$ dependent on $s$. This dependence is captured by the flow equation, which can be written as

$$\frac{\partial \beta^n(s)}{\partial s} = -\mathrm{sgn}\left(A(s)\right)^n_{i_*},$$  (39)

where the sign of the flux jacobian is obtained by diagonalizing the latter and by taking the sign of its eigenvalues. For interacting integrable systems, from Eq. (36), this implies

$$\mathrm{sgn}(A)^j_i = \int \mathrm{d}\lambda \, h^{\mathrm{dr}}_i(\lambda)\,\mathrm{sgn}(v^{\mathrm{eff}}(\lambda))\, h^j_{\mathrm{dr}}(\lambda).$$  (40)

By inserting Eq. (39) into Eq. (3) and by taking the derivative with respect to $s$, the flow equation can be recast in a form where the dependence of the pseudo-energy $\varepsilon(\lambda;s)$ on $s$ is exposed

$$\frac{\partial \varepsilon(\lambda;s)}{\partial s} = -\mathrm{sgn}(v^{\mathrm{eff}}(\lambda;s))h^{\mathrm{dr}}_{i_*}(\lambda;s).$$  (41)

Notice that, as a consequence of Eq. (39), all the dressed quantities, which depend on the GGE state, acquire an additional dependence on the parameter $s$. Exploiting Eqs. (39), and equivalently (41), the SCGF $G(s)$ can be eventually computed as

$$G(s) = \int_0^s \mathrm{d}s' \langle j_{i_*}(0,0)\rangle_{\vartheta(0,0;s')},$$  (42)

where we stress that the current expectation on the r.h.s. is taken over the homogeneous GGE and it is thereby given by Eq. (15). In Eq. (42) we have extended the notation introduced after (8) for the average over the local, homogeneous GGE $\vartheta(0,0;s')$ at the space-time point $(0,0)$ by including the additional dependence on $s$ because of the biasing of the measure.

Equations (39) (or equivalently Eq. (41)) and (42) are the two main results of Refs. [71,72] and they are the necessary equations to compute $G(s)$. First one has to solve numerically Eq. (41) to fix the dependence of the state on $s$, and then the result must be plugged into Eq. (42) to get $G(s)$. This procedure has been carried on in Ref. [71] for the classical hard-rod gas and for the quantum Lieb-Liniger model, where $G(s)$ has been computed in the homogeneous steady state developing at long times from the partitioning protocol initial state in Eqs. (11) and (12). In the classical hard-rod fluid, moreover, the cumulants obtained from the series expansion of $G(s)$ according to Eq. (33) have been compared against Monte-Carlo simulations, finding an excellent agreement and thus corroborating the validity of the approach. In concluding this Subsection, we mention that the flow equation in Eq. (39) can be proved, as shown in Ref. [72], solely on the basis of linear fluctuating hydrodynamics and hydrodynamics projections [11,90] without the need of using any tool coming from integrability and Bethe ansatz. Only in deriving the form in Eq. (41) the TBA machinery is exploited. The expression for $G(s)$ in Eq. (42) together with Eq. (39) is therefore expected to apply more generically to any system, integrable or not integrable, displaying ballistic transport.

## 4 The Euler-scale SCGF for inhomogeneous states

This Section contains the main result of this work, which is the exact expression of $G(s,x,t)$ in the Euler-scaling limit $z \to \infty$ as per Eq. (32), with the initial state $\rho_0$ the inhomogeneous GGE in Eq. (9). In Subsec. 4.1 we state for the reader's convenience the result and the equations

necessary for the numerical computation of $G(s, x, t)$. In Subsec. 4.2 we report the derivation of the results. In Subsec. 4.3 the results for the first three cumulants of the time-integrated current, according to Eq. (33), are reported. In Subsec. 4.4 the non-interacting limit of our general expression is studied.

## 4.1 SCGF for inhomogeneous GGEs: statement of the result

For the *inhomogeneous* and *non-stationary* GGEs $\rho_0$, as in Eq. (9), the evaluation of $G(s, x, t)$ is much more difficult and has not been considered before. In this Section we provide the first expression for the SCGF for inhomogeneous and non-stationary states in the form of $\rho_0$ in (9) in the limit $z \to \infty$ according to Eq. (32).

Consider the inhomogeneous state $\rho_0$ in Eq. (9). Then $G(s, x, t)$ defined in Eq. (32) can be computed as

$$G(s, x, t) = \frac{1}{t} \int_0^s \mathrm{d}s' \int_0^t \mathrm{d}\tau \, \langle j_{i_*}(0, 0) \rangle_{\vartheta(x, \tau; s')}, \tag{43}$$

where the average current is evaluated on the homogeneous GGE $\vartheta(x, \tau; s')$ in Eq. (2) and it is therefore given by Eq. (15). The fundamental difference with respect to the homogeneous case, using the same extension of the notation introduced for Eq. (42), is that the GGE $\vartheta(x, \tau; s')$ depends not only on the parameter $s'$ as a consequence of the biasing of the measure in Eq. (9) via the exponential of the time-integrated current, but also on the (scaled) space-time coordinates $(x, \tau)$ of the fluid cell, because, for every deformation parameter $s'$, the state is inhomogeneous and non-stationary.

To be accurate, the bias of the measure by the exponential of the time integral of the current as in Eq. (32), depends not only on $s$, but also on the parameters $x, t$ characterising the (scaled) space-time position of the integration interval (see, for instance, Fig. 1). Therefore, an average at the fluid cell $(x', t')$ in the deformed state should be denoted as $\langle \ldots \rangle_{\vartheta(x', t'; x, t, s)}$. For lightness of notation, we keep implicit the $x, t$ dependence of the bias itself in the fluid-cell average notation; these can be considered as fixed parameters throughout. The dependence on $s$ is important, as for instance this is integrated over in Eq. (43).

As in the homogeneous case, the $s$ dependence is described by a flow equation for an $s$-dependent state, which in the inhomogeneous case is however significantly more complex than Eq. (39). In terms of the fluid-cell Lagrange parameters described by $\beta^n(x', t'; s)$, it takes the following form:

$$\frac{\partial \beta^n(x', t'; s)}{\partial s} = -\int_0^t \mathrm{d}\tau \, \mathrm{d}\lambda \left[ \Gamma_{(x, \tau) \to (x', t')} V^{j_{i_*}}(x, \tau; s) \right](\lambda) \, h_{\mathrm{dr}}^n(x', t', \lambda; s), \tag{44}$$

where the propagator $\Gamma_{(x, \tau) \to (x', t')}$ has been defined in Eq. (28) with $\tau$ as initial time according to the discussion after Eq. (29). In Eq. (44) we are further using the integral-operator notation introduced in Eq. (27). The one-particle form factor $V^{j_{i_*}}$ of the current $j_{i_*}$ is given in Eq. (24).

It is convenient to re-write this as a flow equation for the pseudo-energy $\varepsilon$. This directly follows upon differentiating the left and the right hand side of Eq. (3) with respect to $s$

$$
\begin{aligned}
\frac{\partial \varepsilon(x', t', \lambda; s)}{\partial s} &= \sum_n \frac{\partial \beta^n(x', t'; s)}{\partial s} h_n(\lambda) + \int \mathrm{d}\mu \, T(\lambda, \mu) \vartheta(x', t', \mu; s) \frac{\partial \varepsilon(x', t', \mu; s)}{\partial s} \\
&= \sum_n \frac{\partial \beta^n(x', t'; s)}{\partial s} \left( h_n(\lambda) + \int \mathrm{d}\mu \, T(\lambda, \mu) \vartheta(x', t', \mu; s) h_n^{\mathrm{dr}}(x', t', \mu; s) \right) \\
&= \sum_n \frac{\partial \beta^n(x', t'; s)}{\partial s} h_n^{\mathrm{dr}}(x', t', \lambda; s),
\end{aligned} \tag{45}
$$

where in the first line we have used the chain rule and the relation $\vartheta(\varepsilon) = \mathrm{d}F(\varepsilon)/\mathrm{d}\varepsilon$. In the second line we have again used the chain rule and the identity $h_n^{\mathrm{dr}}(\lambda) = \partial\varepsilon/\partial\beta^n$, which follows upon differentiating with respect to $\beta^n$ Eq. (3) and by recognizing the integral equation in Eq. (6) defining the dressing operation. Upon inserting Eq. (44) into the last equality in Eq. (45) one obtains the flow equation for the pseudo-energy $\varepsilon$:

$$\frac{\partial\varepsilon(x',t',\lambda;s)}{\partial s} = -\int_0^t \mathrm{d}\tau \left[\Gamma_{(x,\tau)\to(x',t')}V^{j_{i_*}}(x,\tau;s)\right](\lambda), \qquad (46)$$

where we have used the completeness relation in Eq. (37). Using the decomposition of the propagator $\Gamma$ in Eq. (28), the last equation can be written as

$$\frac{\partial\varepsilon(x',t',\lambda;s)}{\partial s} = -\sum_{\delta\in\tau_\star(x',t',\lambda,x)} \mathrm{sgn}(v^{\mathrm{eff}}(x,\delta,\lambda;s))h_{i_*}^{\mathrm{dr}}(x,\delta,\lambda;s)$$
$$-\int_0^t \mathrm{d}\tau\left[\Delta_{(x,\tau)\to(x',t')}V^{j_{i_*}}(x,\tau;s)\right](\lambda), \qquad (47)$$

where we have defined the set of times $\tau_\star(x',t',\lambda,x) = \{\tau : \mathcal{U}(x',t',\lambda,\tau) = x\}$, with the characterstic curve defined in Eqs. (18) and (19).

Notice that for the homogeneous GGE $\vartheta$, the indirect propagator $\Delta$ vanishes, as detailed after Eq. (83) in Appendix A. The characteristic curve, from Eq. (19), in this case is simply given by $\mathcal{U}(x',t',\lambda,\tau) = x' - v^{\mathrm{eff}}(\lambda)(t'-\tau)$, the set $\tau_\star$ is composed by one element only and Eq. (47) reduces to Eq. (41). In the same limit, the time-integral in Eq. (43) trivializes, as the current average value is independent on time, and Eq. (42) is re-obtained. One therefore sees that Eqs. (44) and (47) generalize the results of Refs. [71, 72], recalled in Subsec. 3.2, by including non-stationarity and inhomogeneous situations, when motion occurs at the Euler scale of hydrodynamics. Equations (43), (44) and (47) are the main results of this paper.

Equations (43) and (47) involve the propagator $\Gamma_{(x,\tau)\to(x',t')}$ which, as we have seen in Sec. 2.3, describes the motion of the quasi-particles between $(x,\tau)$ and $(x',t')$ in the inhomogeneous fluid background (under the homogeneous evolution Hamiltonian of the model considered). It is consequently clear that the Euler-scale expression for $G(s,x,t)$ is expected to apply in the same limit where the expression for the two-point correlator of Subsec. 2.3 does. In particular, based on the findings of Ref. [37], where the formulas for two-point functions have been tested again numerical simulations of the microscopic hard-rod dynamics, we expect the expression for $G(s,x,t)$ in Eqs. (43) and (47) to apply for smooth initial inhomogeneities, large enough $z$ in Eq. (9), and long times. The actual verification of this statement by comparing the predictions coming from Eqs. (43) and (47) against simulations of the hard-rod gas will be addressed in a future publication, together with the evaluation of $G(s,x,t)$ for some specific quantum models, e.g., the Lieb-Liniger and the sinh-Gordon. In this work, we will solely present the general theory based on Eqs. (43) and (47) describing the large deviation theory of ballistically transported conserved quantities.

We conclude this Subsection by noting that, if the state $\vartheta_0$ is invariant under simultaneous rescaling of space and time $(x,t) \to (ax,at)$, and therefore $\beta^n(ax',at';0) = \beta^n(x',t';0)$ with $a > 0$ an arbitrary positive constant (e.g., the partitioning protocol initial state in Eqs. (11) and (12)), then the expression in Eq. (43) becomes a function of the scaling variable $\xi = x/t$. As a matter of fact, using the property of the propagator $\Gamma$, valid if $\beta^n(ax',at';0) = \beta^n(x',t';0)$ for every $n$, $x'$ and $t'$,

$$\Gamma_{(x,\tau)\to(x',t')} = a\,\Gamma_{(ax,a\tau)\to(ax',at')}, \qquad (48)$$

which directly follows from Eqs. (28) and (81), the following scaling property is obtained for the Lagrange multipliers $\beta^n(x',t';s)$ for arbitrary values of $s$ by using Eq. (48) in Eq. (44)

$$\beta^n(ax',at';s) = \beta^n(x',t';s). \qquad (49)$$

Equation (49) in turn implies that the same scaling property holds also for the pseudo-energy $\varepsilon(x', t', \lambda; s)$ and any dressed quantity $h_i^{\mathrm{dr}}(x', t', \lambda; s)$:

$$\begin{aligned}
\varepsilon(ax', at', \lambda; s) &= \varepsilon(x', t', \lambda; s) \\
h_i^{\mathrm{dr}}(ax', at', \lambda; s) &= h_i^{\mathrm{dr}}(x', t', \lambda; s).
\end{aligned} \tag{50}$$

Exploiting the last identity, the expression for $G(s, x, t)$ can be eventually written in the form

$$G(s, \xi) = \int_0^s \mathrm{d}s' \int_0^1 \mathrm{d}t' \, \langle j_{i_*}(0, 0) \rangle_{\vartheta(\xi, t'; s')}. \tag{51}$$

In the next Subsection, we provide the derivation of the main results Eqs. (43), (44) and (47).

## 4.2 SCGF for inhomogeneous GGEs: derivation of the result

We start by defining the $s$-tilted ensemble as that obtained by biasing the measure of the inhomogeneous GGE in Eq. (9) with the time-integrated current in Eq. (30). This procedure is analogous to the one introduced in Refs. [52–54, 93] for one-dimensional critical systems described by conformal field theory, and later extended in Refs. [62, 73] to non-interacting systems. We will further comment in Subsec. 4.4 about the relation between the present approach and how it simplifies to the case of non-interacting systems.

Averages over the $s$-tilted ensemble will be denoted as $\langle \dots \rangle_{\mathrm{inh}, z}^{(s)}$, with $\langle \dots \rangle_{\mathrm{inh}, z}^{(0)} = \langle \dots \rangle_{\mathrm{inh}, z}$ by construction. Namely, for a local operator $\mathcal{O}(zx', zt')$ at the space-time point $(zx', zt')$ the $s$-tilted ensemble is defined as

$$\langle \mathcal{O}(zx', zt') \rangle_{\mathrm{inh}, z}^{(s)} = \frac{\langle \mathcal{O}(zx', zt') \exp(s \Delta q_{i_*}(zx, zt)) \rangle_{\mathrm{inh}, z}}{\langle \exp(s \Delta q_{i_*}(zx, zt)) \rangle_{\mathrm{inh}, z}}, \tag{52}$$

where $\Delta q_{i_*}$ is given in Eq. (30). From the definition (52) one also has

$$\frac{\partial \langle \mathcal{O}(zx', zt') \rangle_{\mathrm{inh}, z}^{(s)}}{\partial s} = \int_0^t \mathrm{d}\tau \, z \langle \mathcal{O}(zx', zt') j_{i_*}(zx, z\tau) \rangle_{\mathrm{inh}, z}^{c, (s)}, \tag{53}$$

with the integrand on the r.h.s. denoting the two-point connected correlation function over the $s$-tilted ensemble with the notation of Eq. (52).

It is crucial to note that, although the $s$-tilting involves an integral over time, this *does not affect the dynamics*. The bias is to be understood as a modification of the measure, that is, of the distribution of states at time 0, or on any chosen time slice. According to (52), the measure is modified by a weight which is evaluated by evolving the observables in time (or equivalently, evolving the distribution of states in time), and evaluating the exponential of the time-integrated current. The dynamics is still given by the original, homogeneous and time-independent Hamiltonian of the model under consideration, and thus, at the Euler scale, time slices are related to each other by the original GHD equation (17) of the model. This is important, as we can then use, below, the results for correlations in inhomogeneous and dynamical states reviewed in Subsec. 2.3, which, as emphasised, are based on the assumption of an homogeneous and time-independent dynamics.

The definition of the $s$-tilted measure in Eq. (53) is widely known in the context of large deviations in classical stochastic systems, see, e.g., Refs. [94, 95], and in open quantum systems. In the latter framework it is named "s ensemble", see, e.g., Refs. [96, 97]. There, the approach consists in relating the bias in $s$, as an exponential of the time-integrated current, to a modification of the Lindbladian ruling the evolution of the system. This operation can

be done via the so-called "quantum Doob" transformation, as shown in Refs. [98, 99], and it amounts to a biasing of the probability measures which makes rare events typical. The approach we pursue here is complementary to the quantum Doob transform, in the sense that here we relate the insertion of the exponential of the time-integrated current to a change of the (inhomogeneous) GGE measure. Fundamentally, the latter modification still produces a (inhomogeneous) GGE, whose $s$-dependence can be determined exactly in terms of the flow equation in Eqs. (44) and (47).

We now show that the biasing procedure in Eqs. (52) and (53) defines a flow in the manifold of the inhomogeneous GGEs. Since, by construction, the dynamics is unchanged, we may think of the manifold of inhomogenenous GGEs as described by a space-dependent GGE on any given time slice, $\{\beta^n(x', t'; s)\}_{x',n}$. Let us choose the time slice $t' = 0$. One can adapt the argument developed in Ref. [72] for the homogeneous case, summarized in Subsec. 3.2. Namely, one defines the Lie derivative $\mathcal{L}$ at the point $\langle ... \rangle_{\vartheta_0}^{\text{Eul}}$ in the manifold of inhomogeneous GGEs as

$$\mathcal{L}\langle \mathcal{O}(x', 0) \rangle_{\vartheta_0}^{\text{Eul}} = \int_0^t d\tau \, \langle \mathcal{O}(x', 0) j_{i_*}(x, \tau) \rangle_{\vartheta_0}^{c, \text{Eul}}. \tag{54}$$

Using Eq. (26) with $\tau$ as initial time for the propagator $\Gamma_{(x,\tau) \to (x',t')}$, and Eqs. (23) and (24), the Lie derivative can be written as

$$\mathcal{L}\langle \mathcal{O}(x', 0) \rangle_{\vartheta_0}^{\text{Eul}} = -\sum_n \frac{\partial \langle \mathcal{O} \rangle_{\vartheta(x', 0)}}{\partial \beta^n(x', 0)} \int_0^t d\tau \, d\lambda \left[ \Gamma_{(x,\tau) \to (x', 0)} V^{j_{i_*}}(x, \tau) \right](\lambda) h_{\text{dr}}^n(x', 0, \lambda). \tag{55}$$

Equation (55) shows that the Lie derivative lies on the tangent space to the manifold of inhomogeneous GGEs identified by the Lagrange multipliers $\{\beta^n(x', 0)\}_{x',n}$. Comparing Eqs. (54) and (55) with (53), and remembering the definition of the Euler-scaling limit of two-point correlation functions in Eq. (21), one realizes that the $s$-tilted measure defines a flow directed along the Lie derivative as $z \to \infty$. Accordingly, from Eq. (53), infinitesimal $s$ modifications lie on the tangent space to that manifold, and given that at $s = 0$ the state $\langle ... \rangle_{\text{inh},z}^{(0)}$ is an inhomogeneous GGE and lies on that manifold, then it remains so even after the $s$-tilting. As a consequence, one has for the Euler-scaling limit $z \to \infty$ of Eq. (52) that

$$\lim_{z \to \infty} \langle \mathcal{O}(zx', zt') \rangle_{\text{inh},z}^{(s)} = \langle \mathcal{O}(x', t') \rangle_{\vartheta_0(s)}^{\text{Eul}} = \langle \mathcal{O}(0, 0) \rangle_{\vartheta(x', t'; s)}, \tag{56}$$

where in the second equality we have used the local relaxation assumption (8) thereby expressing the average of the local operator $\mathcal{O}$ over the local homogeneous GGE in Eq. (2) at the fluid cell $(x', t'; s)$. It is also worth to remark that in the second equality of (56) we have extended the notation introduced after (21) for the mode-occupation function $\vartheta_0(s)$, corresponding to the state $\rho_0$ in Eq. (9) as $z \to \infty$, by exposing the additional dependence on $s$ due to the biasing of the measure.

In order to fix the additional dependence on $s$ due to the tilting in Eq. (52) one further needs to specify the flow equation. To do this we start from (53) together with Eq. (56) by choosing the local observable $\mathcal{O}(x', t')$ as some conserved density $q_n(x', t')$ of the model

$$\frac{\partial \langle q_n(x', t') \rangle_{\vartheta_0(s)}^{\text{Eul}}}{\partial s} = \int_0^t d\tau \, \langle q_n(x', t') j_{i_*}(x, \tau) \rangle_{\vartheta_0(s)}^{c, \text{Eul}}. \tag{57}$$

The homogeneous GGE at every fluid cell $(x', t')$ is specified by the Lagrange multipliers $\{\beta^n(x', t'; s)\}_{x',n}$ or, equivalently, by the average conserved densities $\langle q_n(x', t') \rangle_{\vartheta_0(s)}^{\text{Eul}} = \langle q_n(0, 0) \rangle_{\vartheta(x', t'; s)}$ in Eq. (14). Equation (57) can therefore be considered as the "equation of motion" of the state coordinates $\langle q_n(0, 0) \rangle_{\vartheta(x', t'; s)}$ for their trajectory,

parametrized by $s$, in the manifold of inhomogeneous GGEs. For convenience, we express these coordinates on an arbitrary time slice $t'$.

As a matter of fact, Eq. (57) relates the tangent vector of the trajectory, the l.h.s., to a function of the coordinate itself, the r.h.s.. Since the expression of the correlator on the r.h.s. is known, from Eq. (29) in Subsec. 2.3, the equation of motion is well defined and it fixes the flow of the inhomogeneous GGE in Eq. (9) as a function of $s$. Indeed we now show that Eq. (57) is equivalent to the flow equation (44). Exploiting the chain rule one has

$$
\frac{\partial \langle q_n(x',t') \rangle^{\mathrm{Eul}}_{\vartheta_0(s)}}{\partial s} = \int \mathrm{d}y \sum_j \frac{\delta \langle q_n(x',t') \rangle^{\mathrm{Eul}}_{\vartheta_0(s)}}{\delta \beta^j(y,t';s)} \frac{\partial \beta^j(y,t';s)}{\partial s}
$$
$$
= -\sum_j (C_{\vartheta(x',t';s)})_{nj} \frac{\partial \beta^j(x',t';s)}{\partial s} = -\left[ \frac{\partial \beta(x',t';s)}{\partial s} C_{\vartheta(x',t')} \right]_n , \qquad (58)
$$

where in the last step we used vector-matrix notation and the symmetry of the correlation matrix $C_{\vartheta(x',t')}$, defined in Eq. (25). Notice that in Eq. (58) we have further exploited the fluctuation dissipation relation in Eq. (20)

$$
-\frac{\delta \langle q_i(x',t') \rangle^{\mathrm{Eul}}_{\vartheta_0(s)}}{\delta \beta^j(y,t';s)} = \langle q_i(x',t') q_j(y,t') \rangle^{\mathrm{c,Eul}}_{\vartheta_0(s)} = \delta(x'-y)(C_{\vartheta(x',t';s)})_{ij} , \qquad (59)
$$

where the second equality comes from the fact that for equal-time connected correlation functions the propagator $\Gamma_{(y,t') \to (x',t')}(\lambda,\mu) = \delta(x'-y)\delta(\lambda-\mu)$. The latter equality simply expresses that at the Euler scale fluid cells on the same time slice separated in space are uncorrelated, as shown in Ref. [30]. The inverse matrix $C^{-1}_{\vartheta(x',t';s)}$ can be defined by means of the functions $h^j_{\mathrm{dr}}$ in Eq. (37) as

$$
(C^{-1}_{\vartheta(x',t';s)})^{jn} = \int \mathrm{d}\lambda \, \rho_p^{-1}(x',t',\lambda;s) f^{-1}(x',t',\lambda;s) h^n_{\mathrm{dr}}(x',t',\lambda;s) h^j_{\mathrm{dr}}(x',t',\lambda;s) . \qquad (60)
$$

Multiplying Eq. (58) by $C^{-1}_{\vartheta(x',t';s)}$ in Eq. (60) and equating with the r.h.s of Eq. (57), where the connected correlator is given by Eq. (29), one eventually obtains the flow equation for the Lagrange parameter $\beta^n(x',t';s)$ in Eq. (44). The flow equation for $\varepsilon(x',t',\lambda;s)$ (46), (47) follows from Eq. (44) according to the steps reported in Eq. (45).

Once the flow equation is established, proving Eq. (43) for the SCGF is simple. By applying a derivative with respect to $s$ to $G(s,x,t)$ in Eq. (32), with $\rho_0$ as given by Eq. (9), one obtains

$$
\frac{\partial G(s,x,t)}{\partial s} = \frac{1}{t} \int_0^t \mathrm{d}\tau \lim_{z \to \infty} \langle j_{i_*}(zx,z\tau) \rangle^{(s)}_{\mathrm{inh},z} = \frac{1}{t} \int_0^t \mathrm{d}\tau \langle j_{i_*}(x,\tau) \rangle^{\mathrm{Eul}}_{\vartheta_0(s)}
$$
$$
= \frac{1}{t} \int_0^t \mathrm{d}\tau \langle j_{i_*}(0,0) \rangle_{\vartheta(x,\tau;s)} , \qquad (61)
$$

where in the first equality we have used the definition in Eq. (52) for the average over the $s$-tilted ensemble, while Eq. (56) has been exploited in the second and third equality. Integrating in $s$, and exploiting the fact that $G(s=0,x,t)=0$, from the definition in Eq. (32) of the SCGF, one can see that Eq. (61) reduces to Eq. (43).

Regarding the numerical evaluation of Eqs. (43) and (47), we remark that they are expressed in terms of quantities available from the TBA for $s=0$, i.e., in the unbiased case, such as $v^{\mathrm{eff}}(x,\delta,\lambda;0)$, $h^{\mathrm{dr}}_{i_*}(x,\delta,\lambda;0)$ and $V^{j_{i_*}}(x,\tau;0)$. These functions can be efficiently evolved in time using the solution of the GHD equation in Eq. (17) via the characteristic curves in Eqs. (18) and (19), as discussed in Ref. [29]. The most difficult element to numerically compute, which,

instead, is not directly provided by the TBA, is the indirect propagator $\Delta_{(x,\tau)\to(x',t')}$. For the calculation of the latter, however, an efficient numerical scheme has been developed in Ref. [37] within the iFluid open-source code [100]. Accordingly, the only step which has not yet been pursued in the literature, so far, is the numerical solution of the flow equation in Eq. (47). This passage can be achieved by writing Eq. (47) as an integral equation (by formally integrating in $s$ the right and the left hand side). The latter can be in turn solved by iteration starting from the knowledge of the right hand side of Eq. (47) for $s = 0$, where all the TBA functions and the propagator $\Delta_{(x,\tau)\to(x',t')}$ are known as just explained. The resulting expression for the pseudo-energy $\varepsilon(x', t', \lambda; s)$ as a function of $s$ can then be plugged into Eq. (43), together with the expression in Eq. (15) for the average current $\langle j_i(0,0)\rangle_{\vartheta(x,\tau;s')}$, to eventually compute $G(s, x, t)$. We plan to carry on this numerical analysis in a future work.

In concluding this Subsection, we emphasize that the present derivation of $G(s, x, t)$ is strongly based on the result of Ref. [30], recalled in Subsec. 2.3, for two-point functions in inhomogeneous and dynamical GGEs. Accordingly, Equations (43), (44) and (47) are restricted to integrable models, differently from the results in Eqs. (39) and (42) for homogeneous GGEs. The calculation of the scaled cumulant generating for inhomogeneous and dynamical states in generic, not necessarily integrable, models is an unexplored and challenging problem which goes beyond the analysis of the present manuscript.

## 4.3 Analysis of the cumulants

We report here the first three cumulants, which can be obtained from the series expansion in Eqs. (32) and (33) of the general expression given by Eqs. (43) and (47). The first cumulants provide important information about the shape of the probability distribution of $J_{i_*} = \Delta q_{i_*}/t$ and they are the easiest to access experimentally. Moreover, the cumulants can be computed numerically, e.g., with Monte-Carlo simulations in classical systems such as the hard-rod gas, and therefore they can be used to test the predictions of the Euler-scale formula in Eqs. (43) and (47) with simulations of the microscopic dynamics. Higher cumulants can be in principle derived as well, even if the derivation becomes combinatorially more cumbersome as the order increases.

The first cumulant is trivial from Eq. (43) and it is just the time integral of the GHD expression of the current in Eq. (15)

$$\left.\frac{\partial G(s, x, t)}{\partial s}\right|_{s=0} = c_1 = \frac{1}{t}\int_0^t \mathrm{d}\tau \int \frac{\mathrm{d}\lambda}{2\pi}(E')^{\mathrm{dr}}(x, \tau\lambda)\vartheta(\lambda, x, \tau)h_{i_*}(\lambda). \tag{62}$$

For higher-order cumulants we need the useful relation, with $X^{\mathrm{dr}}(x', t', \lambda; s)$ a generic dressed quantity (we drop for brevity all the independent variables),

$$\partial_s X^{\mathrm{dr}} = \frac{\partial\varepsilon}{\partial s}f X^{\mathrm{dr}} - \left(\frac{\partial\varepsilon}{\partial s}f X^{\mathrm{dr}}\right)^{\mathrm{dr}}, \tag{63}$$

which directly follows upon differentiating with respect to $s$ Eq. (6). $\partial\varepsilon/\partial s$ is specified by the flow equation (46) and (47). The second cumulant is related to the Gaussianity of the distribution close to the mean value and, from Eqs. (43) and (63), its expression reads

$$c_2 = -\frac{1}{t}\int_0^t \mathrm{d}\tau \int \frac{\mathrm{d}\lambda}{2\pi}v^{\mathrm{eff}}(x, \tau, \lambda)\rho_p(x, \tau, \lambda)\left.\frac{\partial\varepsilon}{\partial s}\right|_{s=0}f(x, \tau, \lambda)h_{i_*}^{\mathrm{dr}}(x, \tau, \lambda)h_{i_*}^{\mathrm{dr}}(x, \tau, \lambda), \tag{64}$$

which, using (46) turns out to be

$$
\begin{aligned}
c_2 &= \frac{1}{t} \int_0^t d\tau \int_0^t d\tau' \, \langle j_{i_*}(x,\tau) j_{i_*}(x,\tau') \rangle_{\vartheta_0}^{c,\text{Eul}}, \\
&= \frac{1}{t} \int_0^t d\tau \int_0^t d\tau' \int d\lambda \, [\Gamma_{(x,\tau') \to (x,\tau)} V^{j_{i_*}}(x,\tau')](\lambda) \rho_p(x,\tau,\lambda) f(x,\tau,\lambda) V^{j_{i_*}}(x,\tau,\lambda), \quad (65)
\end{aligned}
$$

that is in agreement with the expression for the connected current-current two-point correlation function one obtains from Eq. (29), therefore providing an important consistency check of Eqs. (43) and (46). The expression of the third cumulant $c_3$ is related to the asymmetry or skewness of the distribution. Its expression was not known before and it is given by

$$
\begin{aligned}
c_3 &= \frac{1}{t} \int_0^t d\tau \int_0^t d\tau' \int_0^t d\tau' \, \langle j_{i_*}(x,\tau) j_{i_*}(x,\tau') j_{i_*}(x,\tau'') \rangle_{\vartheta_0}^{c,\text{Eul}} \\
&= -\frac{1}{t} \int_0^t d\tau \int d\lambda \, v^{\text{eff}}(x,\tau,\lambda) \rho_p(x,\tau,\lambda) f(x,\tau,\lambda) \times \\
&\quad \times \left[ 2 \frac{\partial \varepsilon}{\partial s}\Big|_{s=0} \left( \frac{\partial \varepsilon}{\partial s}\Big|_{s=0} f h_{i_*}^{\text{dr}} \right)^{\text{dr}} - h_{i_*}^{\text{dr}} \left( \frac{\partial^2 \varepsilon}{\partial s^2}\Big|_{s=0} + \left( \frac{\partial \varepsilon}{\partial s} \right)^2 \Big|_{s=0} \tilde{f} \right) \right], \quad (66)
\end{aligned}
$$

with

$$
\tilde{f} = \frac{df}{d\varepsilon} \frac{1}{f} + f. \quad (67)
$$

We notice that both Eq. (64) for $c_2$ and Eq. (66) for $c_3$ are written in terms of $\partial \varepsilon / \partial s$, which is given by the flow equation (46) and (47). Once the latter is numerically solved, also the cumulants can be therefore evaluated numerically. We note that in the cases where the initial state $\vartheta_0$ is invariant under the rescaling $(x,t) \to (ax,at)$, with $a > 0$ an arbitrary constant, and therefore Eqs. (48)-(51) apply, the cumulants $c_k(\xi)$ in Eq. (33) of the time integrated current are scaling function of $\xi$, i.e., the connected correlation functions $\langle [\Delta q_{i_*}(x,t)]^k \rangle_{\vartheta_0}^{c,\text{Eul}}$ become scaling functions of $\xi$ once they are rescaled by $t$. We have numerically checked, by simulating the hard-rod gas dynamics from an initial step inhomogeneity in the inverse temperature $\beta^2(x,0)$ as in Eqs. (11) and (12) and in Fig. 1, that the first three cumulants $c_k(\xi)$, with $k \leq 3$, of the particle flow ($h_0(\lambda) = 1$) are functions of $\xi$. In this case, we have observed a linear growth as a function of $t$ of the connected correlation functions, $\langle [\Delta q_0(\xi t,t)]^k \rangle_{\text{inh},z}^c$ with $\xi$ fixed and $k \leq 3$, which implies that the cumulants are finite (the same scaling behavior is expected also for higher cumulants with $k > 3$) and the large deviation principle applies, as commented after Eq. (34). In the homogeneous and stationary limit of the results in Eqs. (43), (44) and (47), the same scaling behavior as a function of $t$ for the cumulants was already observed in Ref. [71]. The fineteness of the cumulants in integrable models implies that there is no divergence in the derivatives of $G(s,x,t)$ w.r.t. $s$ and therefore no dynamical phase transition in the time-integrated current $J_{i_*}$ statistics, see, e.g., Refs. [101–103]. The understanding of the precise reason behind the absence of divergences in the $s$-derivatives of $G(s,x,t)$ in integrable systems is still under investigation and it will not be addressed in this manuscript.

## 4.4 The non-interacting limit

In this Subsection we show how to specify the general result in Eqs. (43) and (44) to non-interacting systems. In particular, we show that the flow equation for the Lagrange multipliers simply reduces to a shift as a function of $s$, as already noted in the homogeneous case in Refs. [71, 72]. This property is characteristic of non-interacting systems and can be ascribed to the fact that the Euler equations in Eq. (13) are in this case linear.

The fundamental simplification happening in non-interacting systems is that no dressing is present since the scattering phase $\Theta(\lambda, \mu)$, or equivalently $T(\lambda, \mu)$ in Eq. (3), vanishes identically. The flow equation in Eq. (44) therefore simplifies drastically. Specifically, the propagator $\Gamma_{(x,\tau)\to(x',t')}(\lambda, \mu)$ in Eq. (28) for free models is uniquely given by the homogeneous contribution (see also Appendix A)

$$\Gamma_{(x,\tau)\to(x',t')}(\lambda, \mu) = \frac{1}{|v_g(\lambda)|}\delta\left(\frac{x'-x}{v_g(\lambda)}-(t'-\tau)\right)\delta(\lambda-\mu),\tag{68}$$

where $v^{\text{eff}}(x,t,\lambda) = v_g(\lambda) = dE(\lambda)/d\lambda$ is the group velocity since there is no dressing. We observe that for the calculation of $G(s,x,t)$ in Eq. (43), where the mean current $\langle j_{i_*}(0,0)\rangle_{\vartheta(x,\tau;s')}$ is evaluated on fluid cells at the fixed space point $x$ and times $\tau \in (0,t)$ (see Fig. 1), one has to specialize the propagator to equal space points $x' = x$, i.e., $\Gamma_{(x,\tau)\to(x,t')}(\lambda, \mu)$. Note that in interacting systems the indirect propagator $\Delta_{(x,\tau)\to(x',t')}(\lambda, \mu)$ vanishes as a consequence of the fact that the quasi-particles trajectories do not depend on the state (see also Appendix A). Inserting Eq. (68) into Eq. (44), with $t' - (x'-x)/v_g(\lambda) < t$, one has

$$\frac{\partial \beta^n(x',t';s)}{\partial s} = -\int_0^t d\tau \int d\lambda\, v_g(\lambda)h_{i_*}(\lambda)h^n(\lambda)\delta\left(\frac{x'-x}{v_g(\lambda)}-(t'-\tau)\right)\frac{1}{|v_g(\lambda)|}$$
$$= -\int d\lambda\, \text{sgn}(v_g(\lambda))h_{i_*}(\lambda)h^n(\lambda) = -\text{sgn}(A)_{i_*}^n,\tag{69}$$

with the last step following from (36) without the dressing. For the same reason $A_i^j$ is independent of $s$ and Eq. (69) can be trivially integrated

$$\beta^n(x',t';s) = \beta^n(x',t')-s\,\text{sgn}(A)_{i_*}^n.\tag{70}$$

In order to further simplify Eq. (69) we briefly recall some basic identities about the Lagrange mulipliers $\{\beta^n\}$. As already commented after Eq. (57), the GGE state at every fluid cell $(x',t')$ can be equivalently described in terms of the Lagrange parameters $\{\beta^n(x',t')\}_{x',n}$ or with the densities $\langle q_n(0,0)\rangle_{\vartheta(x',t')}$. As a matter of fact, it simple to see from Eq. (13), using the definition of the matrix $C$ in Eq. (25), that the multipliers satisfy an hydrodynamic equation similar to the one the densities fulfill, as shown in details in Refs. [9,10],

$$\partial_t \beta^n(x',t') + \sum_j \partial_{x'}\beta^j(x',t')A_j^n(x',t') = 0.\tag{71}$$

Since $A_i^j$ in the absence of dressing is independent of space and time, Eq. (71) is linear. Accordingly, one can introduce the normal mode function $\mathcal{N}(x',t',\lambda)$ via a simple linear combination of the $\{\beta^n(x',t')\}_{x',n}$, i.e.,

$$\mathcal{N}(x',t',\lambda) = \sum_n \beta^n(x',t')h_n(\lambda),\tag{72}$$

where $h_n$ are the bare single-particle eigenvalues, see Eq. (4). Plugging Eq. (72) into Eq. (71) and exploiting the eigenvalue equation for $A_i^j$ in Eq. (38) one has

$$\partial_{t'}\mathcal{N}(x',t',\lambda) + v_g(\lambda)\partial_{x'}\mathcal{N}(x',t',\lambda) = 0.\tag{73}$$

Note that Eq. (73) generalizes to the interacting case upon replacing $v_g(k) \to v^{\text{eff}}(x',t',\lambda)$. As a matter of fact, one can notice that Eq. (73) has the very same structure as Eq. (17). Indeed, the function $\vartheta(x',t',\lambda)$ is the normal mode function associated to the conserved densities $\langle q_n(0,0)\rangle_{\vartheta(x',t',\lambda)}$, as already commented after Eq. (17), in the same way as $\mathcal{N}$ is the mode

function of the Lagrange parameters $\{\beta^n\}$. The fundamental point, specific of free systems, is that $\mathcal{N}$ is *linearly* related to the $\{\beta^n\}$, according to Eq. (72). In the case of interactions, as a consequence of the nonlinearity of Eq. (17), the simple relation in Eq. (73) does not hold, see Refs. [9, 10, 76] for a complete discussion. Inserting Eq. (72) into Eq. (70) and remembering Eq. (38) one has

$$\mathcal{N}(x', t', \lambda; s) = \mathcal{N}(x', t', \lambda) - s\, \text{sgn}(v_g(\lambda)) h_{i_*}(\lambda). \tag{74}$$

Let us now specialize the discussion to the partitioning protocol initial inhomogeneous state in Eqs. (11) and (12) with an initial thermal inhomogeneity in $\beta^2(x', 0)$ and all the other Lagrange parameters initially set to zero (see, for example, Fig. 1). For non-interacting models this protocol has been addressed, e.g., in Refs. [62, 63, 65, 73, 104–109]. In this case, the solution of Eq. (73) is readily found to be

$$\mathcal{N}(\xi', \lambda) = \mathcal{N}_l(\lambda)\Theta(v_g(\lambda) - \xi') + \mathcal{N}_r(\lambda)\Theta(\xi' - v_g(\lambda)), \tag{75}$$

with $\xi' = x'/t'$, while $\mathcal{N}_{l,r}(\lambda)$ are fixed by the initial conditions for the left and right half of the system. Then, from Eq. (72) computed at time $t' = 0$, one has

$$\mathcal{N}(x', 0, \lambda) = \beta^2(x', 0)h_2(\lambda) = \beta_r\, h_2(\lambda)\Theta(x') + \beta_l\, h_2(\lambda)\Theta(-x'), \tag{76}$$

where we identified

$$\mathcal{N}_r(\lambda) = \beta_r\, h_2(\lambda), \qquad \mathcal{N}_l(\lambda) = \beta_l\, h_2(\lambda). \tag{77}$$

Note that $i_* = 2$ in Eq. (74) and $h_2(\lambda) = E(\lambda)$ since we are now considering energy transport. Inserting Eqs. (75) and (77) into Eq. (74) and diving both sides by $h_2(\lambda)$ one eventually has

$$\beta(\xi', \lambda; s) = \beta(\xi', \lambda) - s\, \text{sgn}(v_g(\lambda)), \tag{78}$$

where we defined $\beta(\xi', \lambda) = \mathcal{N}(\xi', \lambda)/h_2(\lambda)$. We observe that for the partitioning protocol initial state $\beta(\xi', \lambda)$ depends on $x'$ and $t'$ through the scaling variable $\xi' = x'/t'$, and therefore the Lagrange multipliers are invariant upon simultaneous rescaling of space and time $\beta^n(ax', at') = \beta^n(x', t')$ for every $n$, $x'$ and $t'$. Accordingly, Eq. (49) is satisfied, as one can explicitly see from Eq. (78), and $G(s, x, t) = G(s, \xi)$, with $\xi = x/t$ according to the discussion before Eq. (51). Upon using Eq. (78) into Eq. (51) one obtains

$$G(s, \xi) = \int_0^s ds' \int_0^1 dt' \int \frac{d\lambda}{2\pi} E(\lambda) v_g(\lambda) \vartheta(\xi/t', \lambda; s'), \tag{79}$$

with

$$\vartheta(\xi/t', \lambda; s) = \vartheta_{\beta_r(s)}(\lambda)\Theta(\xi/t' - v_g(\lambda)) + \vartheta_{\beta_l(s)}(\lambda)\Theta(v_g(\lambda) - \xi/t'), \tag{80}$$

for the filling function $\vartheta$ similarly to Eq. (75), and $\beta_{r,l}(s) = \beta_{r,l} - s\, \text{sgn}(v_g(\lambda))$. For non-interacting systems, accordingly, the biasing of the measure necessary to compute the SCGF can be simply obtained by performing a linear shift of the inverse temperatures $\beta_{l,r}$ characterizing the partitioning protocol initial condition in Eq. (76). The latter statement in the case of the homogeneous stationary state ensuing from the partitioning protocol initial state, obtained upon setting $\xi = 0$ in Eqs. (79) and (80), was already known as "extended fluctuation relation", first proposed in Ref. [93]. The extended fluctuation relation has been shown to be valid in conformal field theory [53, 54] and in free-particle models [62, 63, 93]. In Ref. [72], in particular, the validity of the extended fluctuation relation has been shown to be a characteristic feature of non-interacting systems, where the statistical properties of the time-integrated current are directly determined by the fluctuations of the initial state, induced by the shift of the Lagrange parameters $\beta_{l,r}$, since the quasi-particle trajectories do not depend on the surrounding state where the quasi-particles propagate. Within this perspective, the result in

Eqs. (79) and (80) constitute a generalization as a function of $\xi$ to inhomogeneous and dynamical GGEs in Eq. (9) of the extended fluctuation relation, in agreement with the results of Ref. [73] (cf., Eqs. (84) and (85) therein). In Ref. [73], indeed, Eqs. (79) and (80) have been first derived using stationary phase methods for the partitioning protocol initial state with a thermal inhomogeneity in $\beta^2(x,0)$, as in Eq. (76). The derivation of Eq. (74) presented in this Subsection is, instead, based on Euler-scale hydrodynamics and applies more generally to any initial state $\rho_0$ having the form in Eq. (9). Only in Eqs. (75)-(80) the result has been specialized to the partitioning protocol state with a step profile of $\beta^2(x,0)$. More importantly, the Euler-scale hydrodynamics allows for the exact treatment of interactions, as we have seen in Secs. 4.1 and 4.2. In the case of interacting systems, however, quasi-particles trajectories, due to the nonlinearity of the GHD equation in Eq. (17), are modified by the dynamical inhomogeneous background within which they move. Accordingly, fluctuations do not depend directly on the initial-state fluctuations, but also on the whole motion of the quasi-particles between time 0 and $t$ through the inhomogeneous fluid background, as expressed by the time integral in Eqs. (46) and (47).

## 5  Conclusion

In this manuscript we have addressed the problem of computing the Euler-scaling limit of the scaled cumulant generating function $G(s,x,t)$, defined in Eq. (32) and depicted in Fig. 1, of the time-integrated currents associated to ballistically transported conserved densities in integrable models. The SCGF is of paramount importance since it gives access via the large-deviation principle in Eq. (31) to the full probability distribution $p(J_{i_*})$ of the rare-atypical fluctuation of the rescaled time-integrated current $J_{i_*} = \Delta q_{i_*}(x,t)/t$ far from its mean value $\langle J_{i_*} \rangle$. Moreover, its series expansion provides all the cumulants of the time-integrated current according to Eq. (33). We have extended the analysis of Refs. [71,72], which we have reviewed in Subsec. 3.2, for the calculation of the SCGF in homogeneous and stationary GGEs in Eq. (2) to the more complex and interesting case of *inhomogeneous* and dynamical GGE states of the form in Eq. (9), where we have defined the Euler-scaling limit of $G(s,x,t)$ in Eq. (32).

Our findings are detailed in Sec. 4. Specifically, in Subsec. 4.1 we have stated our main results given by Eqs. (43), (46) and (47), whose proof is provided in Subsec. 4.2. Equation (43) expresses the SCGF as an integral of the mean value of the current, given by the generalized hydrodynamics formula in Eq. (15). The crucial point is that the mean value is computed over an $s$-tilted inhomogeneous measure, see Eqs. (52) and (53), according the exponential of the time-integrated current. All quantities which depend on the inhomogeneous fluid state acquire, as a consequence, an additional dependence on the parameter $s$, coupled to the time integral of the current. This $s$-dependence can be determined by solving the flow equation, cf., Eqs. (46) and (47), which describes exactly how the initial measure is affected by the tilting procedure in terms of a flow, parametrized by $s$, in the manifold of inhomogeneous GGEs. The flow equation is based on the knowledge of the Euler-scale two-point correlation functions, whose expressions have been first derived in Ref. [30] and later numerically tested in Ref. [37] for the hard-rod gas. In Subsec. 4.3 we have provided in Eqs. (62), (64)-(66) the expressions of the first three cumulants of the time-integrated current. We have further numerically checked for the hard-rod fluid, in an initial inhomogeneous state with an inverse temperature $\beta^2(x,0)$ profile as in Eqs. (11) and (12) and in Fig. 1, that the first three cumulants $c_k$ of the particle current, with $k \leq 3$, are finite (higher cumulants are similarly expected to be finite). The numerical check of the finiteness of the cumulants is methodologically fundamental since it confirms, a posteriori, that the time integrals in Eqs. (62), (64)-(66) for the cumulants are convergent, that the series expansion in Eqs. (32) and (33) is well defined and

that therefore the large deviation principle in Eq. (31) applies. In Subsec. 4.4 we have analyzed the non-interacting limit of the general formulas in Eqs. (43) and (44). In this case, due to the linearity of the hydrodynamic equations, the flow equation drastically simplifies and, in the case of the partitioning protocol initial state in Eq. (11), it reduces to a shift of the Lagrange parameters characterizing the initial state, see Eq. (78). We have further checked the the results obtained in this way reproduce those obtained in Ref. [73] via stationary-phase methods for non-interacting systems. In the latter case, the fluctuations of the time-integrated current are directly determined by fluctuations of the initial state since quasi-particles propagate along straight line characteristics with a velocity which is independent on the state. Due to the inhomogeneity and the interactions, on the contrary, quasi-particles move along curved trajectories whose shape depends on the surrounding dynamical fluid state. Consequently, the fluctuations of the time-integrated current depend on the whole trajectories of the quasi-particles, as expressed by Eqs. (46) and (47).

As a future perspective, we plan to numerically evaluate the general expressions in Eqs. (43), (46) and (47) for specific classical and quantum interacting integrable models, such as the Lieb-Liniger, the hard rods, and the sinh Gordon. The comparison between the expressions in Eqs. (62), (64)-(66) and the numerical simulations of the hard-rod model will be also carried out. From the analytical side it would be very interesting to extend our results in Sec. 4 beyond the Euler-scaling limit of infinite variation lengths of the inhomogeneities. This can be achieved by accounting for diffusive corrections to the hydrodynamic equation in Eq. (17), as done in Refs. [43, 86–89]. In the specific case of classical models, such as the hard-rod gas where diffusion has been investigated in detail in Refs. [11, 14, 75], this should somehow connect to the macroscopic fluctuation theory, see Refs. [110–112], which is also based on diffusive hydrodynamics. We surely plan to carry out this study in the future.

## Acknowledgements

G.P. thanks A. Gambassi for careful reading of the manuscript and for collaboration on a related project and M. Kormos for useful comments. The authors thank F. S. Møller for fruitful discussions and collaboration on a related project.

**Funding information**   G.P. thanks the A. Della Riccia Foundation (Florence, Italy)–INFN– for financial support and King's College (London, United Kingdom) for hospitality in the period when this work has been carried on.

## A   Indirect propagator for two-point correlation functions

In this Appendix we report for completeness the necessary formulas for the evaluation of the indirect propagator $\Delta_{(y,0)\rightarrow(x,t)}$ defined in Subsec. 2.3, which enters in the calculation of $G(s, x, t)$ in Eqs. (43) and (47).

The indirect propagator, as shown in Ref. [30], satisfies the integral equation

$$\left[\Delta_{(y,0)\rightarrow(x,t)}V^{\mathcal{O}'}\right](x,t,\lambda) = 2\pi\mathcal{D}(\mathcal{U}(x,t,\lambda,0),\lambda)\Bigg(\left[W_{(y,0)\rightarrow(x,t)}V^{\mathcal{O}'}\right](\lambda)+$$
$$+ \int_{x_0}^{x}\mathrm{d}z\left(\rho_s(z,t)f(z,t)\left[\Delta_{(y,0)\rightarrow(z,t)}V^{\mathcal{O}'}(z,t)\right]\right)^{*\mathrm{dr}}(\lambda)\Bigg), \quad (81)$$

with $V^{\mathcal{O}'}$ defined in Eq. (24) and the characteristic curve in Eqs. (18) and (19). The function $\mathcal{D}$ is usually named "effective acceleration". The latter has been first defined in Ref. [26] to

account for weakly inhomogeneous terms in the Hamiltonian ruling the time evolution of the system, e.g., due to the presence of a confining potential. In the case of Eq. (81), in a complementary way, it expresses the inhomogeneity of the initial state in Eq. (9), while the dynamics is assumed to be dictated by an homogeneous and time-independent Hamiltonian (as commented at the beginning of Subsec. 2.3). The expression of the effective acceleration is given by [26]

$$\mathcal{D}(x,\lambda) = \frac{\partial_x \vartheta(x,0,\lambda)}{2\pi\rho_p(x,0,\lambda)f(x,0,\lambda)} \,. \tag{82}$$

The function $W_{(y,0)\to(x,t)}$ is dubbed source term and it is given by

$$\begin{aligned}
\left[ W_{(y,0)\to(x,t)} V^{\mathcal{O}'} \right](\lambda) = &-\Theta(\mathcal{U}(x,t,\lambda,0)-y)\left( \rho_s(y,0)f(y,0)V^{\mathcal{O}'} \right)^{*\mathrm{dr}}(y,0,\lambda) \\
&+ \int_{x_0}^{x} \mathrm{d}z \sum_{\gamma \in \lambda_\star(z,t,y,0)} \frac{\rho_s(z,t,\gamma)\vartheta(y,0,\gamma)f(y,0,\gamma)}{|\partial_\lambda \mathcal{U}(z,t,\gamma,0)|} T^{\mathrm{dr}}(z,t,\lambda,\gamma)V^{\mathcal{O}'}(\gamma),
\end{aligned} \tag{83}$$

where the star-dressing operation is defined as $h^{*\mathrm{dr}}(\lambda) = h^{\mathrm{dr}}(\lambda) - h(\lambda)$, $\Theta(x)$ is the Heaviside step function, the set $\lambda_\star(x,t,y,0)$ has been defined after Eq. (29), and $x_0$, defined in Eq. (19), has to be fixed such that $\vartheta(x,t',\lambda) = \vartheta(x,0,\lambda)$ for $x < x_0$ and $t' \in [0,t]$, see Refs. [29,30]. In principle one should think that $x_0 = -\infty$, although for numerical calculations, see Ref. [37], one fixes $x_0$ to a value sufficiently far from the point $y$ and any other inhomogeneity. Note that the indirect propagator $\Delta_{(y,0)\to(x,t)}$ vanishes identically in the two limiting cases of an homogeneous initial state, i.e., $\vartheta(x,0,\lambda) = \vartheta(\lambda)$ and therefore $\mathcal{D} = 0$ identically in Eq. (82), and for non-interacting systems, as already commented in Subsec. 4.4 in Eq. (68). In the case of non-interacting models the vanishing of $\Delta_{(y,0)\to(x,t)}$ in Eq. (81) follows from the fact that $h^{*\mathrm{dr}}(\lambda) = h^{\mathrm{dr}}(\lambda) - h(\lambda) = h(\lambda) - h(\lambda) = 0$ for any arbitrary function $h(\lambda)$ since $T = 0$ and there is no dressing. In the two cases of homogeneous initial states and non-interacting systems correlations are then solely given by the direct term in Eqs. (28) and (29).

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
