# Peer review of "Euler-scale dynamical fluctuations in non-equilibrium interacting integrable systems"

_SciPost Physics, doi:SciPost Phys. 10, 116 (2021)_

## Round 1 · Referee Report · Anonymous (Referee 1) · 2021-4-26

Report

This is a deep and important paper. It combines two areas, integrable systems (classical or quantum), specifically generalized hydrodynamics (GHD), and dynamical large deviations. GHD is an extension of traditional hydrodynamics providing a general framework to describe the effective dynamics of integrable systems through the time evolution of the densities and currents of their conserved quantities. It is for dynamics what GGE and TBA are for their statics. This is a fast developing field with Doyon as one of its pioneers.

The second area is that of the large deviation (LD) formalism as applied to dynamics, specifically for the study of the statistics of time-integrated currents in the long-time limit. Dynamical LDs are the natural framework to study rare dynamical events. Most work to date has been on stochastic (classical or quantum) systems. Since the mathematics of LDs is what underpins the standard ensemble method of equilibrium statistical mechanics, that of dynamical large deviations can be thought of as providing a statistical mechanics but for trajectories. This in turn allows to describe dynamics with thermodynamic concepts, such as (dynamical) phases and phase transitions. Dynamical LDs are also closely related to optimal control in relation of how to realise optimally rare dynamical events.

Analytical results in LDs are scarce, especially for interacting systems. This paper provides one such result. This in itself is remarkable. It derives exact expressions (at the Euler scale where GHD applies) for the scaled cumulant generating function (SCGF, the LD function playing the role of a dynamical free-energy) for all time-integrated currents. It does so specifically for a class of non-stationary states, specifically those of the so-called partitioning protocol, and for general integrable systems as described by GHD. The paper generalises an early result by Doyon and others, Refs.71-72, for stationary states (itself an important result).

This is my (elementary) understanding of the above. For systems in a GGE stationary state, the main result of Refs.71-72 was to show that in general within the Euler scale of description the SCGF for a time-integrated current could be written in terms of the average of the instantaneous current in other GGEs, with concrete formulas for how the Lagrange multipliers depend on the counting field (the conjugate of the time-integrated current which is the actual observable quantity). Such an explicit formulation is not present in generic LDs, so this a special property of integrable systems.

This manuscript in turn generalises these earlier results to non-stationary conditions, in particular to the statistics of time-integrated currents following evolution after a quench from a partition-like GGE. This is the second remarkable aspect of the results here: I am not aware of any other analytical derivation of LD functions in many-body systems (classical or quantum) under conditions which are not strictly stationary.

[Studies of LDs away from stationarity have been done before, for stochastic and not integrable systems, for example in the context of growth processes in Klymko et al. Phys. Rev. E 97, 032123 (2018) and Jack Phys. Rev. E 100, 012140 (2019), or for periodcally driven systems in Bertini et al., Ann. Henri Poincaré 19, 3197 (2018). ]

Furthermore, these GHD results (like the earlier ones for stationary conditions) in one go also solve the optimal control problem. The LD functions describe the statistics of rare events through an exponential tilting of the dynamical measure. A related question is what modification to the dynamics is required to generate such rare events in an optimal manner, the so-called Doob transform. Here, since dynamics is ballistic, all fluctuations come from the initial (inhomogeneous) GGE. The solution above for the SCGF immediately solves the Doob transform problem, with the optimal dynamics corresponding to the same dynamics but starting from an appropriately modified inhomogeneous GGE.

The results here will certainly have relevance for GHD more broadly as applied to many-body problems with are difficult to solve by other means. On a much lower level, an exercise that might be illustrative could be to apply them to simpler problems for which the LDs can be solved exactly trough other means. [I am thinking specifically the Rule 54 cellular automaton, a classical deterministic system which is integrable and interacting, and for which GHD is easy to derive (see Buca et al, arXiv:2103.16543) and for which the LDs are solved via MPS, Buca et al. Phys. Rev. E 100, 020103 (2019).]

Overall this a very high standard piece of work. It can be published as is.

---

## Editorial Decision

published